

# The effect of northern forest expansion on evapotranspiration overrides that of a possible physiological water saving response to rising CO₂: Interpretations of movement in Budyko Space

Fernando Jaramillo*[1,2,3], Neil Cory[4], Berit Arheimer[5], Hjalmar Laudon[6], Ype van der Velde[7], Thomas B. Hasper[1], Claudia Teutschbein[8], Johan Uddling[1]

[1]Department of Biological and Environmental Sciences, University of Gothenburg, SE–40530 Gothenburg, Sweden
[2]Department of Physical Geography, Stockholm University, SE–106 91, Stockholm, Sweden
[3]Stockholm Resilience Center, Stockholm University, SE–106 91, Stockholm, Sweden
[4]Department of Forest Resource Management; Division of Forest Resource Data, Swedish University of Agricultural Sciences, Umeå, Sweden
[5]Swedish Meteorological and Hydrological Institute, SE-601 76 Norrköping, Sweden
[6]Department of Forest Ecology and Management, Swedish University of Agricultural Sciences, SE-750 07 Umeå, Sweden
[7]Faculty of Earth and Life Sciences, University of Amsterdam, 1081 HV, Amsterdam, the Netherlands
[8]Department of Earth Sciences, Uppsala University, SE-75236, Uppsala, Sweden.

*Correspondence to*: Fernando Jaramillo (fernando.jaramillo@natgeo.su.se)

**Abstract.** During the last six decades, forest biomass has expanded in the Northern Eurasian basins, mainly due to forest management. This expansion should imply an increasing effect on evapotranspiration. However, increasing global CO₂ emissions also trigger physiological plant water saving responses that induce an opposite effect on evapotranspiration. The
dominant long-term and large-scale effect on evapotranspiration is still a matter of debate. In this study, we determined the dominant effect on evapotranspiration in Northern forests during the period 1961–2012 by studying change-effects on the ratio of actual evapotranspiration to precipitation, known as the evaporative ratio. We used the Budyko framework of water and energy availability at the basin scale to study the hydroclimatic movements in Budyko space of 65 Swedish basins. We found that changes in the evaporative ratio in 60% of these basins could not be explained by climatic changes in precipitation and
potential evapotranspiration. In both the temperate and boreal basin groups studied, a positive residual effect on the evaporative ratio counteracted the negative climatic effect. Furthermore, temporal change of this residual effect during the period 1961–2012 agreed with that of the standing forest biomass in both the temperate and boreal basin groups as well as with that of the forest cover area in the temperate group. Hence, our long-term and regional-scale results indicate that a positive effect on evapotranspiration from the increasing forest biomass overrode any possible negative stomatal water saving response from
increasing atmospheric carbon dioxide concentration. Thus, we suggest that forest expansion is the dominant driver of long-term and large-scale evapotranspiration changes in Northern forests.

## 1 Introduction

Boreal and temperate forests provide important ecosystem services at local, regional and global scales. These services include water regulation, soil stabilisation, biodiversity preservation, and the provisioning of timber, fiber, fuel, food and cultural
values for humans (Chopra, 2005). Changes to the attributes of these forests, such as forest coverage, have had important implications for the functioning of the Earth's climate and water and carbon systems (Abbott et al., 2016; Sterling et al., 2013). The return flow of water vapor from the earth's surface to the atmosphere, known as evapotranspiration, is a key hydroclimatic variable linking these systems. Major regulators of forest evapotranspiration include forest biomass (Feng et al., 2016), leaf stomatal conductance (Lin et al., 2015), canopy leaf area index (LAI) (Mu et al., 2013), tree hydraulic traits (Gao et al., 2014)
and stand surface roughness (Donohue et al., 2007).



However, the dominant effects of forest change on evapotranspiration are still a matter of debate. For instance, Betts et al. (2007) argue for a dominant global decrease of evapotranspiration due to a water saving responses to increasing carbon dioxide ($CO_2$) concentrations that decreases plant stomatal conductance. On the contrary, Piao et al. (2007) argued that this

physiological effect was cancelled by simultaneous $CO_2$-induced increases in plant growth and total canopy leaf area, and that changes in climate and land use were the dominant drivers of temporal changes in evapotranspiration. In the context of the northern latitude, changes in forest attributes are likely to play an important role, because forests in these regions have a higher proportion of available energy partitioned into latent heat than other ecosystems in these regions, such as grassland, wetlands and tundra (Baldocchi et al., 2000; Kasurinen et al., 2014; van der Velde et al., 2013).

Observation-based studies of tree water use responses to elevated $CO_2$ are limited to tree- or plot-scale experiments of a few years (the 17-year study by Tor-ngern et al. (2015) is the longest) due to the difficulty and the costs of performing long-term controlled $CO_2$ experiments in the field. Most of these short-term tree field experiments have found that growth under elevated $CO_2$ has caused a water saving response of decreased stomatal conductance (Ainsworth and Rogers, 2007; Assmann, 1999;

Hasper et al. 2017; Medlyn et al., 2001). However, the water saving response depends on environmental conditions and forest species. For instance, stomatal conductance tends to be less sensitive to high $CO_2$ concentrations in gymnosperms (e.g. conifers) when compared to angiosperms (Hasper et al., 2017; Medlyn et al., 1999). In Northern latitudes, field experiments have reported on leaf water saving responses in Silver birch (Rey and Jarvis, 1998), but not in the coniferous tree species Norway spruce (Hasper et al., 2016). In Scots pine, both lack of effect and a rather small reduction in stomatal conductance

under elevated $CO_2$ have been observed (Kellomäki and Wang, 1996; Sigurdsson et al., 2002)

Tree or plot experiments focusing on the effect of changing forest biomass/cover on evapotranspiration in Northern latitudes are generally related to forest management, since this is the activity responsible for most forest changes (Elmhagen et al., 2015; Laudon et al., 2011). As forest clear-cutting has been found to reduce evapotranspiration and increase runoff, reforestation or

regrowth has resulted in an opposite effect (Andréassian, 2004; Sørensen et al., 2009). For long-term and large-scale studies of this effect, basin-scale hydrological assessments are otherwise required (Andréassian, 2004). Long-term availability of precipitation and runoff data can be used to estimate evapotranspiration changes by water balance over long periods. For example, in Sweden, a basin-scale study in four large basins was used in combination with plot experiments to determine the combined effect of climatic and forest change on forest water use (Hasper et al., 2016). However, further separation of forest

effects from hydroclimatic effects requires the incorporation of more advanced tools, such as the Budyko framework (Budyko, 1974). This framework links the water and energy availability at the basin scale to basin-scale hydrology and has been used to differentiate the hydroclimatic effects of forest conditions and change (Zhang et al., 2001). Such are the cases of studies conducted in Germany that identified the impacts of air quality on forest evapotranspiration (Renner et al., 2013) or in North America to differentiate the controls of water partitioning and the responses of forested basins to climate and human drivers

(Creed et al., 2014; Jones et al., 2012; Wang and Hejazi, 2011; Williams et al., 2012). This framework has also been applied in China and even worldwide to study the hydrological effects of reforestation programs and the forest controls on water partitioning (Fang et al., 2001; Huang et al., 2003; Li et al., 2016; Xu et al., 2014; Zhang et al., 2008b; Zhou et al., 2015)

Our main objective was to determine the dominant forest effect driving changes in the ratio of actual evapotranspiration to

precipitation, known as the evaporative ratio, from a basin scale approach. For this purpose, we used a former framework (Jaramillo and Destouni, 2014, 2015; van der Velde et al., 2014) that studies movement of basins in the space comprising the aridity index and the evaporative ratio, known as the Budyko space. We separated the climatic effect on evapotranspiration that relates to changes in the aridity index from a residual effect that relates to other drivers of change. In forested basins, this residual effect should correspond to that of change in forest attributes. We further explored the potential drivers of this effect

by studying the co-development of the residual effect and that of the forest attributes of standing biomass, forest cover area or species composition. We chose the period 1961–2012 for analyzing the change due to the large availability of runoff and forest attribute data across Sweden during this period.



## 2 Materials and methods

### 2.1 Hydroclimatic data

We gathered data on precipitation ($P$) and runoff ($R$) for 65 unregulated Swedish basins (Fig. 1a). These basins were selected from a group of Swedish basins monitored by the Swedish Hydrologic and Meteorological Institute (SMHI) and compiled by
Arheimer and Lindström (2015). The daily $R$ data for the 65 basin outlets and corresponding basin boundaries were obtained from SMHI's hydrologic server (https://vattenwebb.smhi.se/station/) for the period 1961–2012. The daily $R$ data was aggregated to annual values and we considered only complete years in the analysis, defined as years with at least 98% data capture. We used the annual $P$ and $R$ data to calculate actual evapotranspiration ($E$) as

$$E = P - R - \mathrm{d}S/\mathrm{d}t \tag{1}$$

where $\mathrm{d}S/\mathrm{d}t$ is the change in water storage within the basin. We obtained the daily $P$ data for 68 climatic stations operated by SMHI that had a continuous range of data during the entire period 1961–2012 from SMHI's online server Luftwebb (http://opendata-download-metobs.smhi.se/explore/) and made a spatial interpolation by Thiessen polygon to obtain mean $P$ values for each basin. We performed our hydroclimatic assessments during the period 1961–2012 by calculating the inter-annual means of the two 26-year sub-periods 1961–1986 and 1987–2012 and defined all hydroclimatic and forest changes as
the difference between these means. Using such long-term periods becomes an advantage since when averaged over long periods, $\mathrm{d}S/\mathrm{d}t$ should be considerably smaller than $P$, $R$ and $E$, permitting the basin-scale assumption of essentially zero long-term average water storage change within each subperiod ($\mathrm{d}S/\mathrm{d}t \approx 0$) and between both subperiods. These assumptions are required in order to calculate $E$ by Eq. (1) with available data and the corresponding hydroclimatic changes described in Section 2.2.

We also obtained mean, minimum and maximum daily temperature data ($T$, $T_{\min}$, $T_{\max}$) from the climatic stations with $P$ data for calculations of potential evapotranspiration ($E_0$). Estimates of annual $E_0$ in the 68 stations were obtained by three models: the Langbein model in terms of $T$ (Langbein, 1949), the Hargreaves model in terms of $T_{\min}$ and $T_{\max}$ (Hargreaves et al., 1985), and the FAO Penman-Monteith equation (Allen, 1998). For the latter model, the additional surface wind velocity was obtained from the long-term mean annual geostrophic wind data (i.e., at 1000 m.a.s.l.) modelled by SMHI
(http://www.smhi.se/en/climate/climate-indicators/climate-indicators-geostrophic-wind-1.91478). The $E_0$ station estimates were then interpolated spatially to each basin area by a trivariate spline which allows a spatially varying relationship between $E_0$ and elevation (Tait and Woods, 2007). A fourth estimate of $E_0$ was also obtained for each basin from the Climatic Research Unit-gridded $E_0$ product CRU TS3.23 (Harris et al., 2014). The four $E_0$ estimates to obtain a mean $E_0$ for uncertainty assessment and to minimize the potential problems of relying on a single model (Milly and Dunne, 2016).

### 2.2 Budyko framework

We used the Budyko framework (Budyko, 1974) to characterize the observed partitioning of $P$ into $R$ and $E$ in the 65 basins during the period 1961–2012. The Budyko framework is based on the relationship of the partitioning of water and energy on land and states that evapotranspiration is limited by the supply of either water (i.e., $P$) or energy (i.e., $E_0$). This relationship (Fig. 2) is often represented in the space (i.e., Budyko space) created between the ratio of actual evapotranspiration to
precipitation, known as the evaporative ratio ($\Psi$)

$$\Psi = E/P \tag{2}$$

and the ratio of potential evapotranspiration to precipitation ($\Phi$), known as the aridity index

$$\Phi = E_0/P \tag{3}$$

The relationship between $\Psi$ and $\Phi$ is represented by Budyko-type curves expressing the first in terms of the latter; $\Psi = F(\Phi)$.
The $\Psi$ is expected to increase linearly with $\Phi$ at low values of $\Phi$ (i.e. $E$ is energy limited), while gradually reducing its increasing rate at higher $\Phi$ values (i.e., $E$ is water limited). Here, we use the "Budyko-type" formulation of Yang et al. (2008) that expresses a modelled evaporative ratio ($\Psi_c$) in terms of $\Phi$ and a parameter representing the effect of the contribution of




catchment characteristics, such as vegetation, soils and topography for each basin ($n$). This formulation has the same functional form as that of Choudhury (1999) and has been synthetized by Zhang et al. (2015) as

$$\Psi_c = (1 + \Phi^{-n})^{-1/n} \tag{4}$$

For this study, a mean $n$ value was calculated for each basin by solving Eq. (4) with the mean $\Psi$ and $\Phi$ values of the entire period 1961–2012. A basin changing hydroclimatic conditions from a subperiod 1 ($t_1$) to a subperiod 2 ($t_2$) can be represented in Budyko space by a point moving from initial conditions ($t_1$: $\Phi_1$, $\Psi_1$) (Fig. 2). If the movement is only due to change in the aridity index ($\Delta\Phi$), the movement will occur along the corresponding Budyko-type curve to a new point ($t_{2*}$: $\Phi_{2*}$, $\Psi_{2*}$), implying a change in the evaporative ratio, termed henceforth as the climatic effect ($\Delta\Psi_c$). However, a basin will most certainly move in time due to combination of change in the aridity index and other drivers of change (Gudmundsson et al., 2016; Jaramillo and Destouni, 2014). Under such realistic circumstances, the basin will move to a new location not falling on the initial Budyko-type curve ($t_2$: $\Phi_2$, $\Psi_2$) implying a corresponding observed total change in the evaporative ratio ($\Delta\Psi$).

Regarding the two forest-related effects pertaining to this study, expanding forests should increase $E$ due to an increase in root depth and total canopy leaf area, which, under constant conditions of precipitation, would imply an upward movement in Budyko space (Fig. 2b) under constant $P$. In a similar way, a tree water saving response to increasing atmospheric $CO_2$ concentrations reduces stomata conductance and should decrease ET, implying a downward movement in Budyko space under constant $P$.

Following van der Velde et al. (2014), we represented movement due to changes in the aridity index change as a vector ($\boldsymbol{v^*}$) with direction of movement ($\theta$) and magnitude ($r$) calculated as

$$\theta = b - \arctan\left(\frac{\Delta\Psi_c}{\Delta\Phi}\right) \tag{5}$$

$$r = \sqrt{(\Delta\Psi_c)^2 + (\Delta\Phi)^2} \tag{6}$$

where b is a constant and $\theta$ is in degrees ($0º<\theta<360º$) starting clockwise from the upper vertical (b = 90° when $\Delta\Phi >0$ and b = 270° when $\Delta\Phi<0$). In a similar way, we represent the total movement vector ($\boldsymbol{v}$) based on $R$ and P observations by replacing $\Delta\Psi_c$ with $\Delta\Psi$ in Eqs. 5 and 6.

We used the approach of Jaramillo and Destouni [2014] to synthetize both climate-driven movements and total movements for the 65 basins as typical wind roses of direction and magnitude. It is worth noting that since the magnitude and direction of change in Budyko space depends on the specific hydroclimatic conditions of each basin (Greve et al., 2016; Gudmundsson et al., 2016; Jaramillo and Destouni, 2014) and on the definition of the space in which such movement occurs, such wind roses oversimplify the variability of movements in Budyko space. However, using these wind roses is a simple way to synthetize general tendencies of movement in large sets of basins and enables a first-order identification of the importance of drivers of change different from the aridity index. For instance, directions of movement that go beyond the range of slopes of any Budyko-type curve (i.e., $45º<\theta<90º$ and $225º<\theta<270º$) imply that drivers different from the aridity index dominate the observed changes in the evaporative ratio. This, of course, when the $n$ parameter in Eq. (4) is held constant in time.

In order to identify possible forest-related effects on the evaporative ratio, we separated the non-climatic residual effect from the climatic effect. The residual effect on the evaporative ratio ($\Delta\Psi_r$) was calculated as

$$\Delta\Psi_r = \Delta\Psi - \Delta\Psi_c \tag{7}$$

However, recent findings have shown that changes in the ratio of precipitation in the form of snow to total precipitation also have an effect on $\Psi$ (Berghuijs et al., 2014). Other studies also highlight the effect of seasonality and/or intra-annual changes in precipitation and temperature (Chen et al., 2013; Milly, 1993; Zanardo et al., 2012). For example, shorter but more intensive rain events between prolonged drought periods may decrease the annual $\Psi$ under constant $P$. As such, we also include the possible effects of these drivers in the uncertainty section under Discussion.



## 2.3 Linking the residual effect ΔΨ$_r$ to forest change

The amount in which the residual ΔΨ$_r$ represents forest-related effects should be determined for each basin by comparing change in forest attributes with ΔΨ$_r$. Nevertheless, although the forest attribute data of the Swedish National Forest Inventory (NFI) (http://www.slu.se/en/Collaborative-Centres-and-Projects/the-swedish-national-forest-inventory/) is available during the period 1961–2012, the NFI is designed to provide a reasonably accurate temporal data at the Swedish county level. Since most of the studied Swedish basins are smaller than the counties in which they are located, we opted for spatially aggregating the ΔΨ$_r$ data to basin groups per biome (i.e., temperate and boreal), according to a terrestrial ecoregion classification (Olson et al., 2001) to have a comparable representation of hydroclimatic and forest changes. According to the NFI data, in both basin groups more than 50% of the area is covered by forests (Fig. 1b). We also extracted from the NFI representative and historical forest-attribute data on forest standing biomass (V), forest cover area (A) and leaf area index (LAI) for the area covered by the two basin-groups per biome.

In order to study the temporal co-development of A, V, LAI and ΔΨ$_r$, we calculated a time-step five-year moving average of ΔΨ$_r$, defined for each year j, as

$$\Delta\Psi_{rj} = \Psi_{j+1} - \Psi_j - \Psi_{c\ j+1} + \Psi_{c\ j} \tag{8}$$

and computed the corresponding cumulative ΔΨ$_{r\ j}$, i.e., [ΔΨ$_r$], for direct comparison with the development of the forest attributes.

## 2.4 Forest attributes

According to recent data from NFI, productive forestland now accounts for 57% of the Swedish land surface after constantly expanding throughout the 20[th] century (Nilsson et al., 2016). The productive forest standing volume has increased by 85% since the first NFI took place in 1923 (KSLA, 2015). The data on forest attributes used in this study were collected by the NFI, which utilizes a stratified systematic sample based upon clustered sample plots, designed to deliver statistics at the county level. The inventory quantifies the surface area of all types of land use (i.e., productive forestland, mires, mountainous regions, agricultural land, rock surfaces and urban areas); however, the most comprehensive data is collected for forestland. The inventory is based on a moving five-year sampling average; a detailed description of the inventory design and methods is available in Fridman et al. [2014]. Forest A and V data were extracted for the period 1961–2012 (or 1963–2010 as a five-year moving average) for each of the temperate and boreal basin groups and into four species/species groups: Norway spruce (*Picea abies*), Scots pine (*Pinus sylvestris L.*), Silver birch (*Betula pendula*) and other deciduous broadleaf species in the basin area.

Moreover, we calculated the development of total leaf area index (LAI) of trees in each biome and the fraction of LAI that belonged to coniferous and deciduous tree species, as birch leaves have approximately twice as high transpiration rates per unit projected leaf area compared to Norway spruce and Scots pine needles (Tarvainen et al., 2013; Uddling et al., 2005). We based the LAI quantification on estimates of leaf dry biomass and leaf mass per unit projected leaf area (LMA). Dry leaf biomass for coniferous species ($M_c$; pine and spruce) and deciduous species ($M_d$; birch) were calculated from NFI data and the allometric biomass equations of Marklund (1988) and Repola (2008) (using equations for birch), respectively. Since $M_c$ and $M_d$ could only be calculated from biomass data per species available in the NFI since 1973, total LAI change in each biome was calculated for the period 1973–2012. The total LAI was calculated as the sum of LAI from both coniferous (LAI$_c$) and deciduous forests (LAI$_d$) in each biome basin group as:

$$LAI = LAI_c + LAI_d = \frac{1}{A}\left(\frac{M_c}{LMA_c} + \frac{M_d}{LMA_d}\right) \tag{9}$$

where LMA of spruce and pine (LMA$_c$) is taken as 240 g/m$^2$ (Sigurdsson et al., 2002) and of birch (LMA$_d$) as 71 g/m$^2$ (Härkönen et al., 2015). We used the ratio of deciduous LAI to total LAI (LAI$_Q$) as proxy of the temporal development of tree species composition of the forests. We assumed a conservative sampling standard error of A (5.4%), V (10%) following (Fridman et al., 2014) and a corresponding propagated error for LAI$_Q$.





## 3 Results

### 3.1 Movement in Budyko space and effect separation

Most of the 65 basins presented energy-limited conditions since their aridity index $\Phi$ fell below one (Fig. 3). This means that ET in these basins was more limited by energy demand than by water availability. The aridity indexes $\Phi$ and evaporative ratios
$\Psi$ of the boreal basins were generally lower than those of the temperate basins due to their northerly and cool location. As such, boreal basins have less energy available per unit of precipitation and precipitation is partitioned more into runoff than in temperate basins. Some basins plotted low in Budyko space, i.e., evaporative ratio $\Psi$ near zero, possibly due to underestimates of precipitation due to precipitation undercatch in rain gauges due to falling snow and/or wind, unrealistic measurements of runoff or significant groundwater flux across the basin boundary.

The spectra of movements in Budyko space showed that from the period 1961–1986 to the period 1987–2012, all basins in temperate and boreal biomes experienced a decrease in the aridity index $\Phi$ ($180^\circ < \theta < 360^\circ$; Fig. 4a, c). These general decreases in the aridity index appeared to be the result of an increase in precipitation between the two time periods that are considerably larger than the increase in potential evapotranspiration $E_0$ (Fig. 5). The increase in $P$ occurred mostly around winter (January and February) and late spring – summer (May – August) and was larger in the boreal than in the temperate group of basins.
For instance, the maximum increase of $P$ in the boreal group occurred in June by 25 mm/yr. On the contrary, the increase in potential evapotranspiration $E_0$ was small, with the highest value occurring in April in both biomes (7 mm/yr across biomes).

The spectra of movements in Budyko space between the periods 1961–1986 and 1987–2012 evidenced an increase in the evaporative ratio in most temperate (60%) and boreal (61%) basins, moving them upwards in Budyko space (Fig. 4a, c). However, in the absence of other drivers of change, a decrease in the aridity index can only result in a decrease in the
evaporative ratio (Fig 4b, d). Hence, the upward movements ($270^\circ < \theta < 360^\circ$) that occur along with decreases in the aridity index suggest the influence of other driving effects different from the aridity index. As such, changes in the aridity index are not the only or the most important drivers. The separation of climatic and residual effects shows that, in general, the evaporative ratio in these basins have experienced a decreasing climatic effect driven by less arid conditions ($\Delta\Psi_c<0$) and an increasing residual effect due to other different drivers of change ($\Delta\Psi_r>0$) (Fig. 6). Note that the distribution of change for all basins shows that
this counteracting effect applies to the median, arithmetic and area-weighted means and interquartile ranges of both basin groups. Furthermore, even though evaporative ratio estimates may depend on the model being used (Wang et al., 2015), the residual effect $\Delta\Psi_r$ was positive regardless of the model used to calculate potential evapotranspiration (Fig. 7).

### 3.2 Forest change and its co-development with $\Delta\Psi_r$

The positive residual effect $\Delta\Psi_r$ in both biomes might result from expanding forests, as $E$ likely increases with forest expansion.
Indeed, forests have expanded in terms of volume and partly in terms of area (Fig. 8). Between the two 25-year periods, the forest cover area $A$ and the standing forest biomass $V$ increased by 5% and 38% in the group of temperate basins (Fig. 8). In the group of boreal basins, $V$ also increased by 23%, but $A$ remained stable (a 3% decrease smaller than the sampling standard error). The temperate and boreal forest composition $LAI_Q$ remained stable, based on the non-significant changes between these two periods.

In general, the overall increase in the five-year moving average of the cumulative residual effect $[\Delta\Psi_r]$ was more in agreement in both basin groups with the steady increase in $V$ over the period 1961–2012 than with the change of $A$ or $LAI_Q$ (Fig. 9). While the long-term change of $[\Delta\Psi_r]$ mainly followed that of $V$ in both basin groups, it appeared to follow only that of $A$ in temperate basins. Although the general change of $[\Delta\Psi_r]$ appeared to follow the same inter-decadal oscillation pattern of $LAI_Q$, their relation could not be corroborated due to large estimate uncertainty.




## 4 Discussion

### 4.1 Dominant forest-related effect on evapotranspiration

Our results show that the residual effect of change in the evaporative ratio $\Delta\Psi_r$ increased in almost all basins and forest expansion, mostly in terms of increasing forest biomass, drove this effect in both biomes during the period 1961–2012. An
interpretation of these results is that a positive effect of increasing forest biomass increased the evaporative ratio in these basins. For the same amount of annual precipitation, surface water is being gradually partitioned into more evapotranspiration than runoff as forests expand. Under a hypothetical situation with no changes in climate and stomatal conductance, increasing forest biomass (here 23% for boreal basins and 38% in temperate basins) should increase transpiration if canopy leaf area index LAI and root biomass increase as well. At least in the case of LAI, an increase in Sweden over time was indicated by
our calculations as well as both remote sensing and modelled data (Zhu et al., 2016). Several studies during the last decades and in many forest ecosystems across the world have found a similar increase in evapotranspiration and decrease in runoff with reforestation at the plot scale (Andréassian, 2004; Bosch and Hewlett, 1982; Sørensen et al., 2009) and basin scale (Fang et al., 2001; Huang et al., 2003; Li et al., 2016; Xu et al., 2014; Zhang et al., 2008b).

On the other hand, a possible water-saving response caused by reduced stomatal conductance under rising atmospheric $CO_2$
concentrations should have decreased $\Psi_r$ (Ainsworth and Long, 2005). Hence, our results point to an overriding effect of the increasing forest biomass on that of any potential stomatal water saving response under rising atmospheric $CO_2$. Our results thus contradict studies suggesting that $CO_2$-induced stomatal water-saving responses play a dominant effect on the terrestrial water cycle  (Betts et al., 2007; Gedney et al., 2006), but agree with studies of water use responses of trees in Swedish forests. Coniferous species dominate the forest composition in the studied Northern basins (Fig. 8) and the water saving response was
not observed in Swedish field experiments with the dominating coniferous tree species Norway spruce and Scots pine (Hasper et al., 2016; Sigurdsson et al., 2002). Although a water saving response was documented for the most common deciduous tree species Silver birch (Rey and Jarvis, 1998), our results showed that the deciduous forest fraction was small and did not significantly change over time in these basins (Fig. 8).

### 4.2 Uncertainties in attribution of forest effects

Are there any other factors than the forest attributes included in this study affecting evapotranspiration and evaporative ratio in these basins? The areal extent of forest fires and wind-throws in Sweden is small, and such effects should thus have minor influences on large-scale evapotranspiration. Drainage practices used in forestry and agriculture may have played a role in the partitioning of water in Swedish boreal and temperate landscapes, although the magnitude of this effect is uncertain and differs between different drainage techniques (Wesström et al., 2003). Farmers implemented subsurface drainage systems in croplands
before 1960 and open ditches were implemented in forests throughout the 20[th] century, peaking during the 1930s and late 1970s to 1980s. Hence, it is possible that some fraction of the observed residual effect $\Delta\Psi_r$ corresponds to an effect from drainage. However, the extent of functioning drainage in Sweden today is about 5% of the total land area (Berglund, Ö, et al., 2009). Considering this rather low fraction of drained land and that most of it was implemented before the present study period (1961-2012), it is unlikely that the observed temporal trends in [$\Delta\Psi_r$] are related to drainage.

Flow regulation and hydropower development have been shown to increase the evaporative ratio at local to global scales (Destouni et al., 2013; Jaramillo and Destouni, 2015; Levi et al., 2015). However, all the basins used in this study are unregulated (Arheimer and Lindström, 2015). Technological improvements over time in rain gauges to reduce losses of water due to wind, evaporation and snow undercatch may also reflect as a false increase in precipitation and corresponding increase evapotranspiration. Nevertheless, in Sweden, these improvements occurred before the period of the present study; windshields
were introduced in Sweden in 1853, and the last installations were reported in 1935. Even the replacement of the old cans made out of galvanized iron for new aluminum ones occurred around 1958 (Sofokleous, 2016).

The assumption of steady state conditions of the Budyko framework required to maintain the constant water supply limit may also become a potential source of uncertainty for calculations of $\Delta\Psi_r$ (Chen et al., 2013; Greve et al., 2016; Wang and Tang, 2014; Zhang et al., 2008a). Although we have used mean estimates of evapotranspiration for 26 years to reduce the impact of
such assumption ($dS/dt \approx 0$), the lack of data required to calculate $dS/dt$ impedes the quantification of the consequences of such



an assumption. Long-term non-zero dS/dt can potentially arise in a number of situations (Bring et al., 2015). Recent studies have shown that the Budyko-type curves in Budyko space actually change, and in some cases, extend over the limits of water availability under non-steady conditions (Greve et al., 2016; Moussa and Lhomme, 2016). These changes alter the slopes of the Budyko-type curves. However, the studies of Greve et al. (2016) and Moussa and Lhomme (2016) show that even under non-steady conditions (d$S$/d$t\neq0$), the directions represented by the slopes of such curves are within the same range of directions under steady-state conditions (i.e., 45º<θ<90º and 225º<θ<270º). The fact that the upward movements in Budyko space attributed to forest expansion (i.e., directions in the range 270º<θ<360º) are still outside of the range of such curves representing non-steady conditions points even more to the effect of forest expansion on evapotranspiration in these basins.

Intra-annual or seasonal changes of precipitation and temperature may also result in non-steady conditions and influence in $\Delta\Psi_r$. However, the general seasonality pattern of potential evapotranspiration and precipitation in Northern latitudes was persistent in time (Fig. 4). Recent findings have also shown that temporal changes in the ratio of precipitation falling as snow to total precipitation may also affect the evaporative ratio (Berghuijs et al., 2014). For instance, studies in China have found important changes in the evaporative ratio with decreasing snow fraction after assuming that in such latitudes, spring snowmelt does not infiltrate the soil and mainly flows on land as runoff, not being used by plants for their biological processes (Zhang et al., 2015). Nevertheless, this extreme assumption does not apply to the conditions of Northern Scandinavian forests. Laudon et al. (2004, 2007) found that a majority of the snowmelt water infiltrates the soil in boreal Sweden and is available for plants in spring. Our calculations show that the fraction of precipitation falling a snow decreased from the period 1961-1986 to the period 1987-2012 from 0.20 to 0.14 in temperate basins and from 0.45 to 0.43 in boreal basins. The effects of the change in snow fraction on evaporative ratio may then seem not as important in these Swedish basins, however, further research should be conducted to model and quantify such long-term effect in these latitudes.

## 5 Conclusion

We used the Budyko framework to study movements in Budyko space of 65 unregulated Swedish basins, aiming to detect the dominating hydroclimatic effect of forest change in the Northern forests; either an increase in actual evapotranspiration due to forest expansion or a decrease by plant water saving responses due to rising atmospheric $CO_2$. We found that from the period 1961–1986 to the period 1987–2012, forest biomass increased in both temperate and boreal basin groups, and forest cover area increased only in the temperate basin group. Forest composition did not show any major changes between these two periods. In all basins, the aridity index decreased due to an increase in precipitation larger than a corresponding increase in potential evapotranspiration. This decrease was accompanied in 60% of these basins by an increase in the evaporative ratio, which as the Budyko framework of water and energy availability shows, cannot be explained by the climatic effect of changing aridity index. In general, the observed change-effects in the evaporative ratio are results of the combination of the negative climatic effect and a positive residual effect from other drivers. Temporal development of forest change and the residual effect indicate that the increasing residual effect is due to expanding forests, mostly manifested by increasing forest biomass. Hence, an interpretation of our results points to a positive effect on evapotranspiration of increasing forest biomass that overrides any potential decreasing water saving response under rising atmospheric $CO_2$ due to reductions in stomatal conductance. These results contradict studies suggesting that $CO_2$-induced stomatal water-saving responses have a dominant effect on the water cycle in Northern forests and highlight the implications of expanding Northern forests for the freshwater system at these latitudes.

## Competing interests

The authors declare that they have no conflict of interest.



**Acknowledgements**

The strategic research area Biodiversity and Ecosystem Services in a Changing Climate BECC of Lund University and the University of Gothenburg (http://www.becc.lu.se/), the Swedish Research Council (VR, project 2015-06503) and the Swedish Research Council for Environment, Agricultural Sciences and Spatial Planning (942-2015-740) have funded this study. All
data required to conduct this study will be available in a server of the Physical Geography Department at Stockholm University.

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




**Figure captions**

**Figure 1: Location, land cover and forest change in the 60 basins used in this study. (a)** The location of boreal (green) and temperate (purple) basins, **(b)** Land cover in each biome group of basins in terms of relative area.

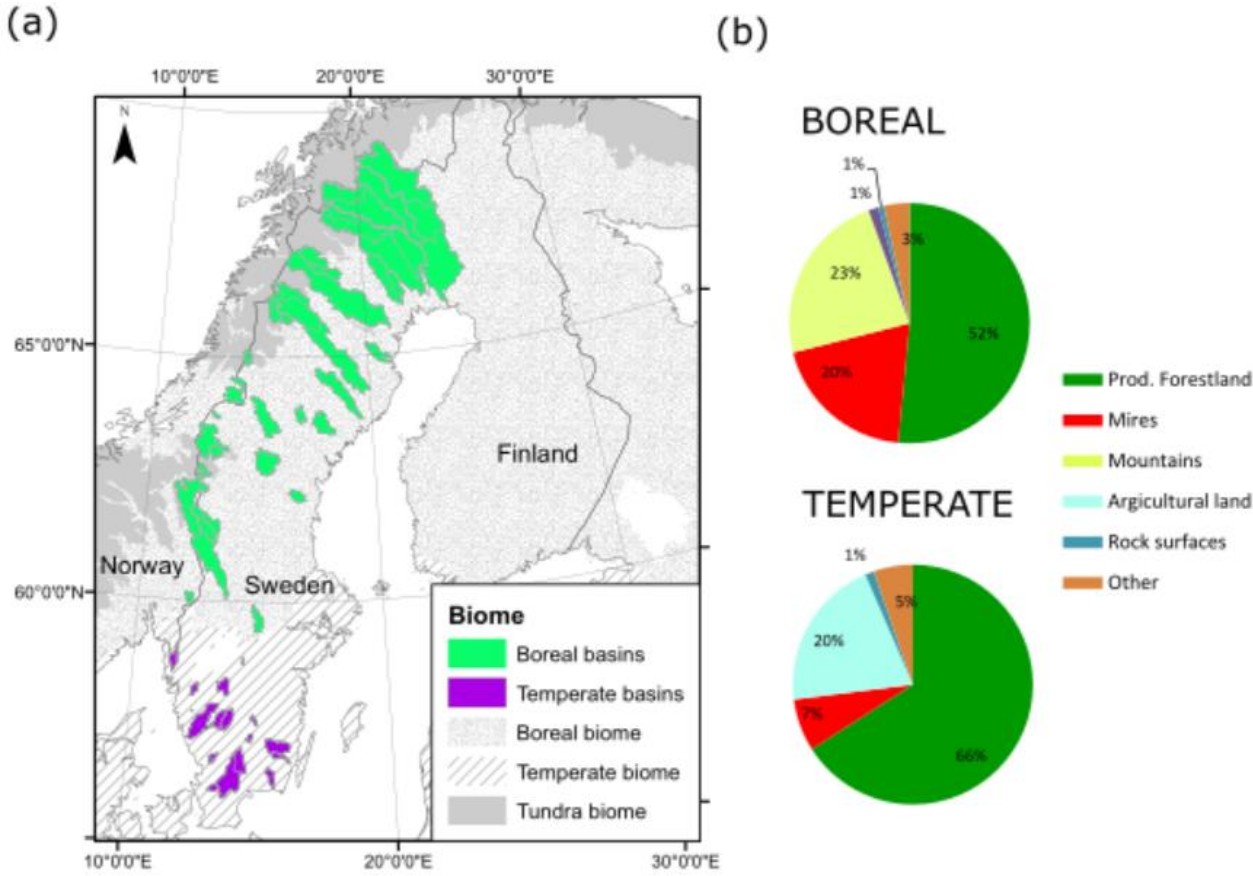



**Figure 2: Schematic representation of movement in Budyko space. (a)** Evaporative ratio ($\Psi = E/P$) vs. aridity index ($\Phi = E_0/P$) and the parameters describing movement in such space from period 1 ($t_1$) to period 2 ($t_1$) by a vector ($v$). The $t_1$ and $t_2$ represent mean conditions of $\Phi$ and $\Psi$ during the periods 1961–1986 and 1987–2012, respectively. The change in evaporative ratio ($\Delta\Psi$) is divided into an effect from changes in the aridity index ($\Delta\Psi_c$), termed the climatic effect and represented by the vector ($v^*$), and an effect from other drivers, termed the residual effect ($\Delta\Psi_r$). **(b)** Expected directions of movements in Budyko space with fixed $\Phi$) of the effects from expanding forests (increase in forest biomass, forest cover area and/or LAI) and water saving response to increasing atmospheric $CO_2$ concentrations.

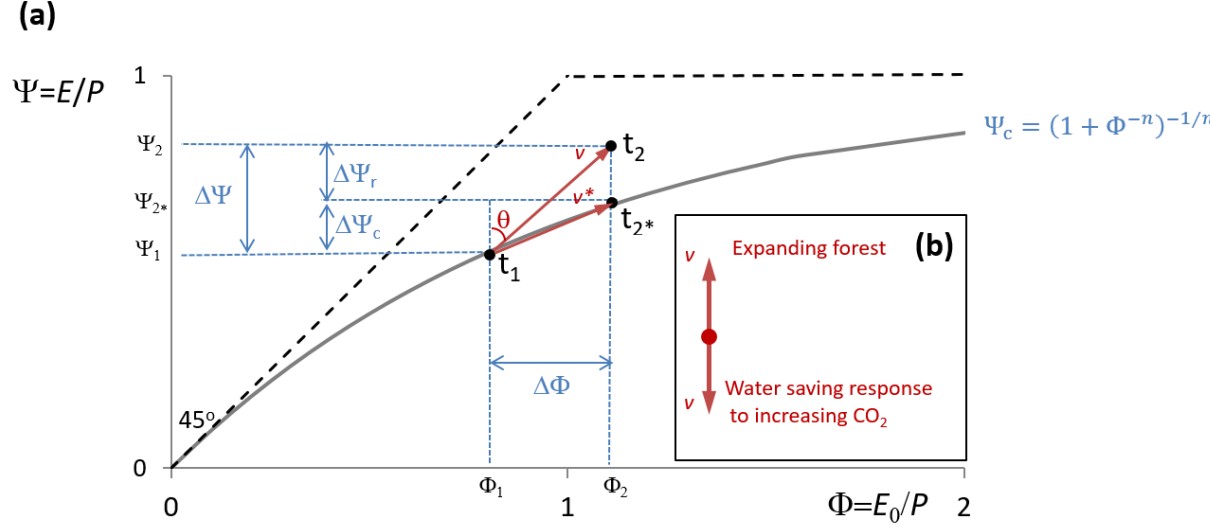





**Figure 3: Hydroclimatic conditions in Budyko space. (a)** Mean hydroclimatic conditions of the 60 basins during the period 1961–2012 illustrated in Budyko space, in terms of the aridity index (Φ; x-axis) and evaporative ratio (Ψ; y-axis) for temperate (purple) and boreal (green) basins.

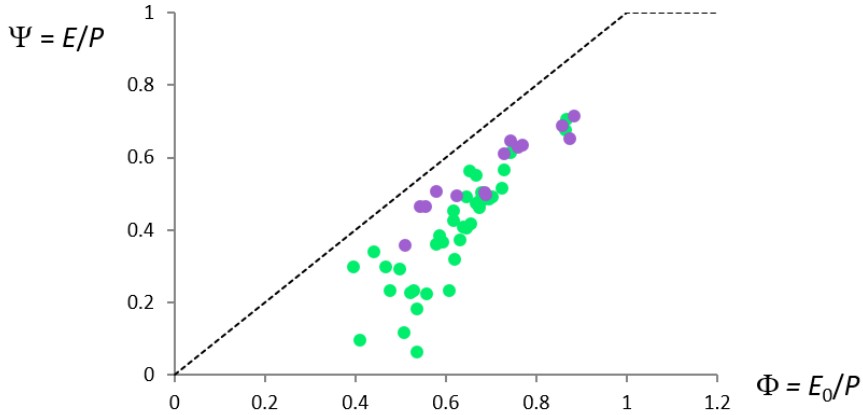




**Figure 4: Hydroclimatic movement in Budyko space due to changes in aridity index (Φ) and evaporative ratio (Ψ) between the two comparative periods 1961–1986 and 1987–2012.** Wind roses of individual basin movements of the boreal basin-group according to: **(a)** the combined effect of all drivers of change by calculating Ψ from runoff observations (i.e., Ψ by Eq. (2)) and **(b)** the effect of only change in the aridity index by calculating Ψ from modeled data (Ψ$_c$ by Eq. (4)), with fixed n and varying Φ). **(c–d)** Similar wind roses for temperate basins. The range of directions of movement (0<θ<360°) is divided into 15° interval-paddles that group all basins moving in each direction interval. Intensity of color intervals represent the magnitude of the movements (*r*) in Budyko space in such given direction θ.

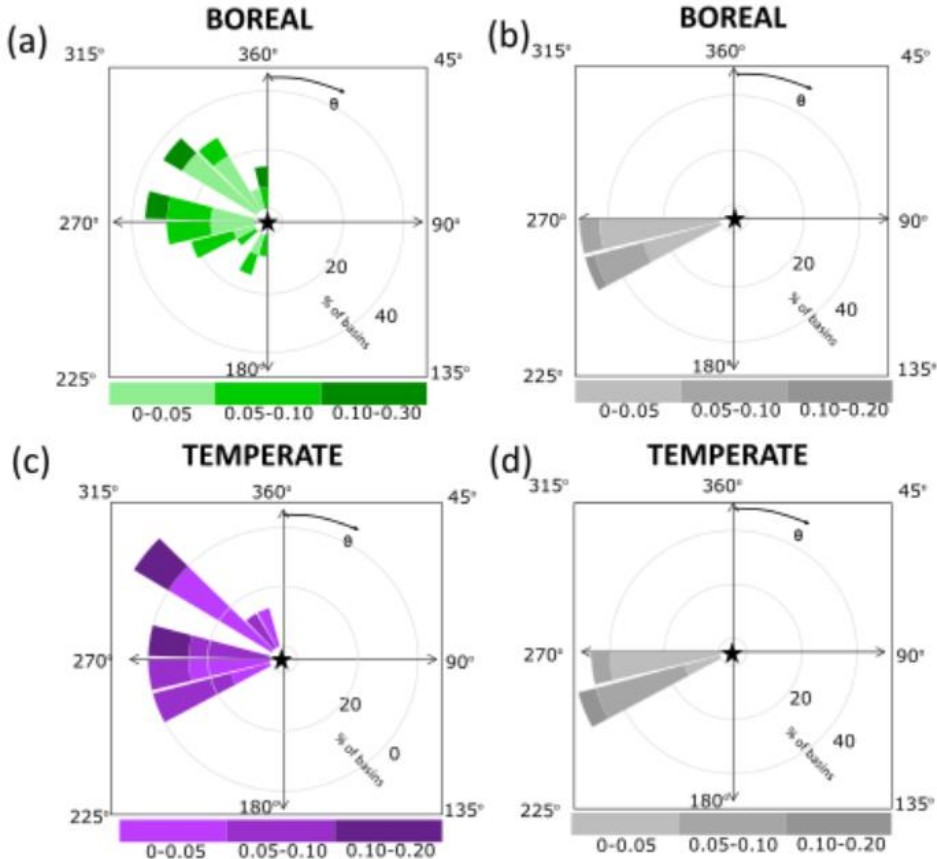





**Figure 5: Intra-annual change in precipitation ($P$) and potential evapotranspiration ($E_0$).** Changes in the mean of $P$ and $E_0$ between the periods 1961–1986 and 1987–2012 in the boreal **(a–b)** and temperate **(c–d)** basin groups for each month (January (1) to December (12)).

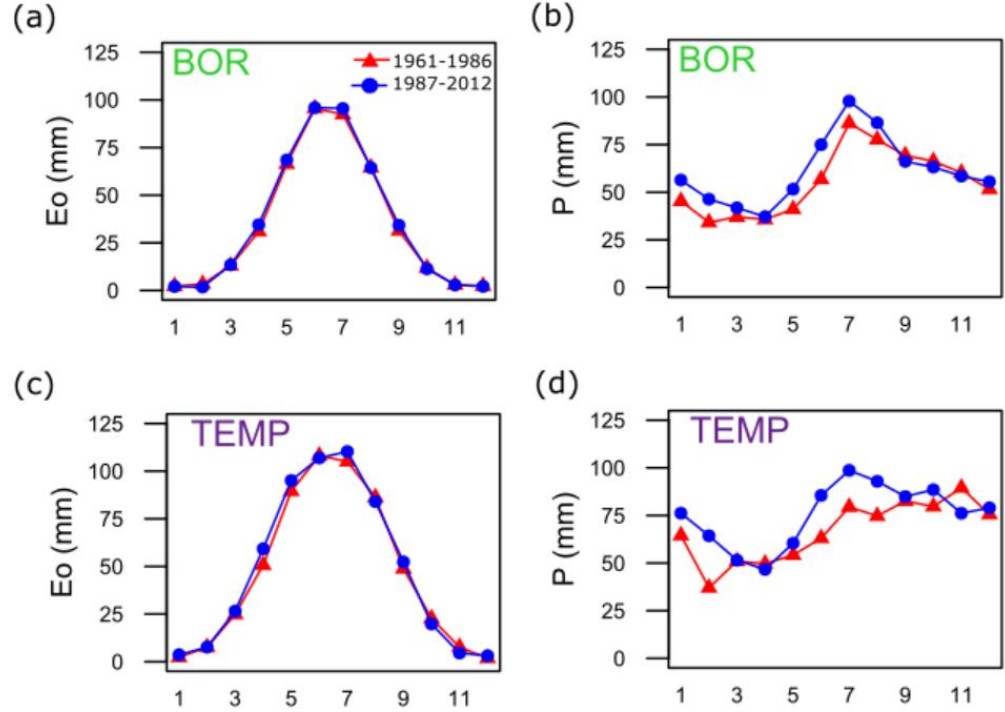





**Figure 6: Climate and residual effects on the evaporative ratio.** Distribution of changes from 1961–1986 to 1987–2012 in the evaporative ratio ($\Delta\Psi$) and its climatic ($\Delta\Psi_c$) and residual ($\Delta\Psi_r$) effect-components in the **(a)** boreal and **(b)** temperate groups of basins. Boxplot statistics include arithmetic mean (blue triangles), area-weighted mean (pink circles), median (thick horizontal black line), interquartile range (IQR) (boxes), whiskers (confidence interval of $\pm1.58*\text{IQR}\sqrt{N}$ where $N$ is the number of basins in each biome group) and outliers (small black circles).

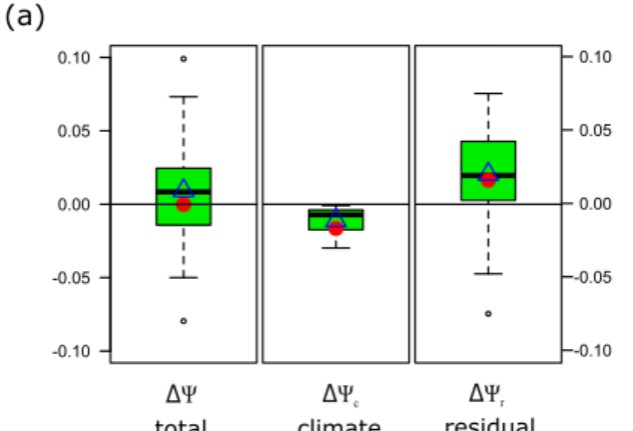

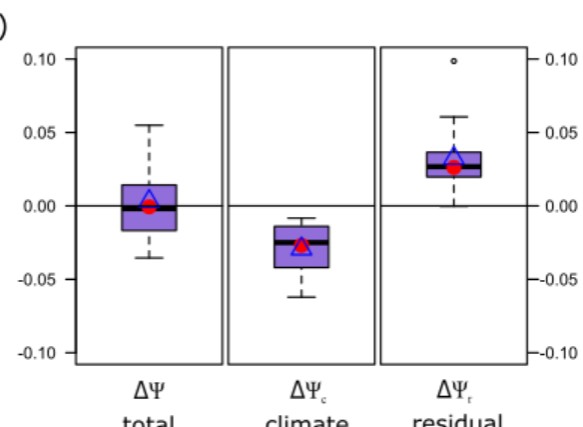





**Figure 7: Uncertainty from potential evapotranspiration models.** Distribution for all 65 basins of the residual effects on the evaporative ratio ($\Delta\Psi_r$) by calculating potential evapotranspiration with four different models/products: **(a)** CRU 3.23 Potential Evapotranspiration grid product, **(b)** Hargreaves, **(c)** FAO Penman-Monteith and **(d)** Langbein.

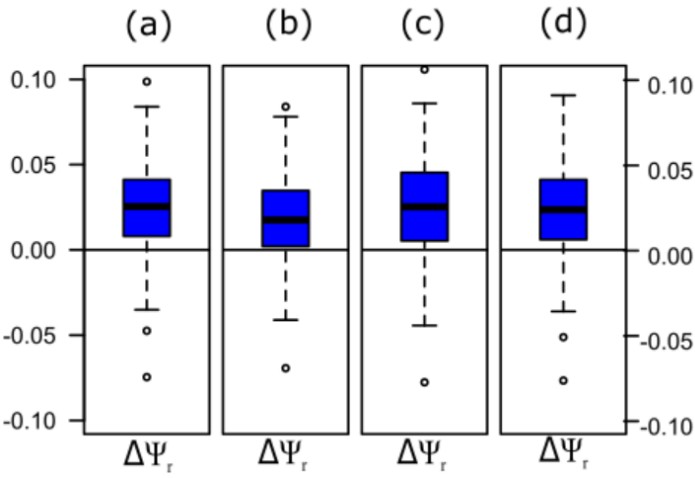



**Figure 8: Changes in forest attributes per biome.** Change calculated as the percentage change between the comparative 25-year periods 1961–1986 and 1987–2012 of forest cover (*A*), forest standing biomass (*V*) and change in forest composition, represented as the ratio of deciduous forests to total leaf area index (LAI$_Q$). We assume a conservative sampling standard error of *A* (5.4%), *V* (10%) and a corresponding propagated error for LAI.

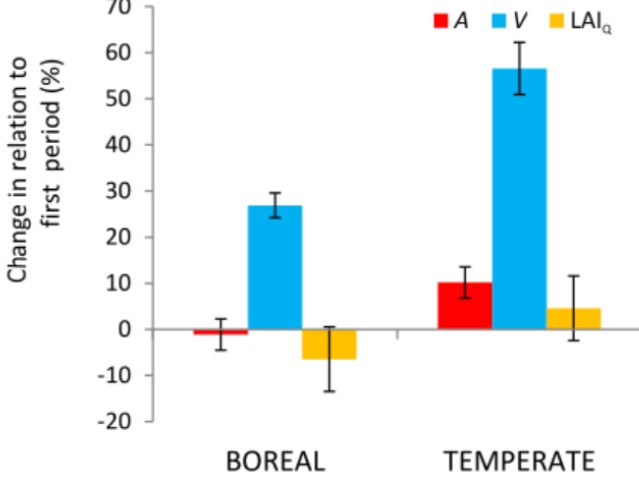





**Figure 9: Co-evolution of the cumulative residual effect on the evaporative ratio ($[\Delta\Psi_r]$) and forest development.** Five-year moving window on the year-by-year area-weighted development of $[\Delta\Psi_r]$ during the period 1961–2012 (black; left-axis) and forest attributes (standing volume $V$, forest cover $A$ and change in species composition $LAI_Q$; right axis) for **(a)** boreal and **(b)** temperate biomes.

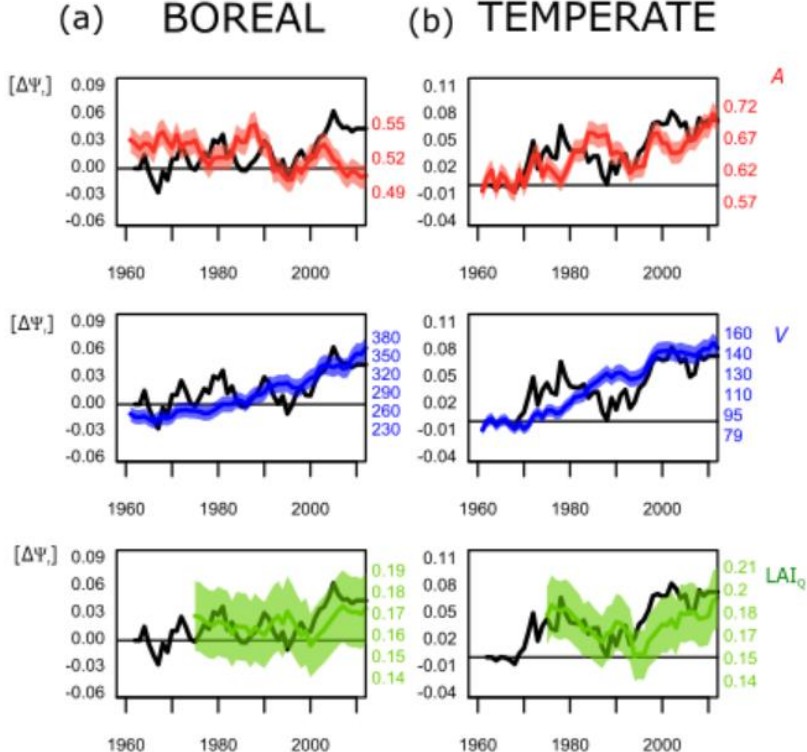