# Peer review of "Dominant effect of increasing forest biomass on evapotranspiration: Interpretations of movement in Budyko Space"

_Hydrology and Earth System Sciences, 2017_

## Referee Comment (RC1) · Anonymous Referee #1 · 17 Jul 2017

Overview This manuscript addresses causes of water balance changes in Swedish forests during the period 1961 to 2012. Water balance changes were encoded in estimates of the evaporative fraction, i.e., annual actual evapotranspiration over precipitation (E/P). The estimated changes in E/P were explained by climatic changes due to precipitation and potential evapotranspiration and by ecosystem changes due to standing forest biomass. The authors conclude that E/P increased 1961-2012 and attribute this increase to increased forest cover, despite a concurrent increase in precipitation

(i.e., decrease in the aridity index). The authors anticipated a reduction in E/P due to $CO_2$ fertilization and interpret observed increased E/P as evidence that increased forest area overcompensated for any $CO_2$ fertilization effect.

The authors address a question of broad interest with relevance to water and carbon budgets from watershed to global scales. As noted by the authors, recent high profile papers have addressed the $CO_2$ fertilization effect on the water balance directly (Betts et al. 2007) and in conjunction with climate, land use, and leaf area (Piao et al. 2007). As another example, Swann et al. (2016) and Milly and Dunne (2016) both suggest $CO_2$ fertilization-induced decreases in ET may partially mitigate currently projected changes in continental drying and drought severity. The current paper follows in the footsteps of Piao et al. (2007) by addressing the effect of reforestation on basin-scale evapotranspiration in the context of simultaneous $CO_2$ and associated climatic changes.

In general, the hypothesis and analysis were well thought out and executed. Broadly, I think the clarity of the paper could be improved. And, more specifically, I have some difficulty understanding the analysis linking evapotranspiration changes to forest expansion rather than $CO_2$ fertilization.

"Possible physiological water saving response to rising $CO_2$" The authors state in their title that the ET increase from forest expansion overrides ET decrease from rising $CO_2$. The authors then present evidence that (1) the aridity index decreased over time due to an increase in precipitation; (2) this decrease in aridity index lead to a decrease in E/P, as expected from the Budyko curve; and (3) there was an overall increase in E/P. This increase in E/P was then attributed to changes in forest standing biomass. There was no evidence or estimate of the $CO_2$ fertilization effect and, therefore, I find it difficult to conclude that this effect was present and indeed over-compensated.

To address this issue, I would suggest the authors reduce the focus on $CO_2$ fertilization in the title and abstract. My intuition is that the $CO_2$ fertilization effect was relatively

weak in this ecosystem. The atmospheric CO2 concentration increased approximately 85 ppm (315 too 400 ppm) over the study period, 1961-2012. In a meta-analysis of the FACE experiments (Ainsworth and Rogers 2007), trees showed one of the lowest responses of stomatal conductance to elevated CO2 (~20% decrease). In these experiments, CO2 was increased from 366 ppm to 567 ppm, an increase 2.35 times that experienced from 1961 to 2012. As a first guess, one might expect less than 10% decrease in stomatal conductance whereas the authors show that forest standing biomass increased 25% and 55% in boreal and temperate watersheds (Figure 8).

Further, in several places, the authors make the same argument based on results of species-specific responses. See page 2, lines 9-20 and page 7, lines 14-23. To paraphrase, the watersheds studied are dominated by coniferous species and CO2 water saving response has not been observed in these species.

In conclusion, there is little evidence to expect a CO2 water saving response in the studied watersheds. Therefore, I do not think it is appropriate to set up the paper with this hypothesis that is later not supported with the data. On the other hand, I do think it is appropriate to address this weak CO2 effect as one reason why an increase in E/P was observed, as the authors have done on page 7.

Specific comments

Hydroclimatic Data:

1) The temporal scales of the data are inconsistent. For the Penman-Monteith model, the long-term mean annual geostrophic wind at 1000 meters above sea level is used. Given fine-scale variability in windspeed and its local, leaf-scale effect on transpiration, this approach is not warranted. This is especially true for comparison with the Langbein and Hargreaves models, which use daily temperature as model input. My suggestion is to remove the Penman-Monteith analysis and use the 3 other sources for PET.

2) The discussion in the first paragraph surrounding equation (1) is disorganized. My

suggestion is to place the description of the P data before equation (1). Then follow with the sentence, "We used annual P and R data to calculate ..." after you've described what the annual P and R data are.

Budyko Framework:

1) I am not familiar with the Psi notation for evaporative index and typically see it written as E/P or something similar. It may help the reader to be consistent with notation from your references.

Linking the residual effect Delta Psi_r to forest change:

1) I don't understand equation 8. It is described as a five-year moving average, but there is only a j and j+1 term – does this mean only the current and previous year are used in the calculation?

Figure 2: Can you re-orient the arrow from t1 to t2 to be consistent with your result of decreasing aridity index? That would make the figure easier to read.

Figure 3: Can you include a separate Budyko plot for the early and late time periods? That would give the reader a general, more intuitive sense of how the watersheds moved in the Budyko space.

Figure 6: What is the y-axis label in this figure? Same for Figure 7.

References

Ainsworth, E.A. and Rogers, A., 2007. The response of photosynthesis and stomatal conductance to rising [CO2]: mechanisms and environmental interactions. Plant, cell & environment, 30(3), pp.258-270.

Betts, R. A., Boucher, O., Collins, M., Cox, P. M., Falloon, P. D., Gedney, N., Hemming, D. L., Huntingford, C., Jones, C.D., Sexton, D. M. H. and Webb, M. J.: Projected increase in continental runoff due to plant responses to increasing carbon dioxide, Nature, 448(7157), 1037–1041, doi:10.1038/nature06045, 2007.

Milly, P.C. and Dunne, K.A., 2016. Potential evapotranspiration and continental drying. Nature Climate Change, 6, pp.946-949.

Piao, S., Friedlingstein, P., Ciais, P., de Noblet-Ducoudre, N., Labat, D. and Zaehle, S.: Changes in climate and land use have a larger direct impact than rising CO2 on global river runoff trends, Proc. Natl. Acad. Sci. U. S. A., 104(39), 15242– 15247, doi:10.1073/pnas.0707213104, 2007.

Swann, A.L., Hoffman, F.M., Koven, C.D. and Randerson, J.T., 2016. Plant responses to increasing CO2 reduce estimates of climate impacts on drought severity. Proceedings of the National Academy of Sciences, 113(36), pp.10019-10024.

---

## Referee Comment (RC2) · Anonymous Referee #2 · 18 Jul 2017

This manuscript uses climatic (temperature and precipitation), vegetation (forest expansion in Sweden), and runoff time series from 65 unregulated Swedish basins over 1961-2012 to investigate changes in the precipitation partitioning into evapotranspiration (ET) and runoff. The authors are specifically interested in seeing if increase in forest biomass that occurred in the past decades would combine with two competing physiological phenomena to either increase or decrease ET beyond the extent dictated by climate (represented by the aridity index): (1) decrease plant stomatal conductance

in response to increase in CO2 (water saving responses), resulting in a decrease in ET, or (2) CO2-induced increase in plant growth and leaf area, resulting in a increase in ET. The contribution of this manuscript is thus organized into two main components: that of analysis of change, and of attribution of this change to forest properties (total area, volume, and proportion of deciduous species to total LAI).

In my opinion, the authors have made a convincing argument for residual changes in basin-level ET that goes the extent dictated by climate. They have postulated that, because the observed ET has increased despite a decrease in aridity index (when Budyko's curve, under stationary conditions, would suggest otherwise based solely on climatic effects), there must exist some non-climate related mechanisms that offset this increase.

To make this point, however, I think that Figure 4 is redundant with Figure 6. Figure 4's use of "wind roses" does not add additional support for the authors' main point. While they claim that these wind roses are "a simple way to synthetize general tendencies of movement," the general direction of these movements are well summarized by the histograms presented in Figure 6, so to me these are two different graphical representation for very similar sets of information.

In addition, "spectra of movements in Budyko space," used repeatedly in Section 3.1, need to be rephrased. Since "spectra" has a very specific meaning in time series analysis, I would suggest the authors avoid this term in reference to movement in the Budyko coordinates.

I think also that the weakness of the manuscript as it stands lies in formulating the argument for the second point, e.g., in attributing the observed increase in ET to a specific, hypothesized mechanism. In Figure 8, boreal and temperature forests showed opposite changes in this deciduous proportion, though how this might contribute to the overall increase in ET in both forest types is not discussed.

In addition, the relationship between forest attributes and $\Delta\Psi\_r$ is described in Section

3.2 using very vague terms like "in agreement with" and "followed that of." I would suggest applying more statistical analysis (and plot out the correlation between $\Delta\Psi\_r$ and each of the forest attributes) in this section to more quantitatively describe these relationships. I also remain unconvinced of the authors' use of the cumulative $\Delta\Psi\_r$ in comparison to the forest attributes (Figure 9), and the application and choice of a 5-year moving window for $\Delta\Psi\_r$. Both of these usages require further justification.

If the authors can address these concerns, this paper will make a good contribution to the study of water partitioning at high latitudes.

---

## Referee Comment (RC3) · Anonymous Referee #3 · 18 Jul 2017

This paper uses the Budyko framework to study the effect of changes in evaporative ratios at a number of boreal and temperate catchments in Sweden. The study looks at changes in the location of each catchment in Budyko space during two consecutive 25-year periods in the early 21st century and second half of he 20th century, and separates the changes into climatic and non-climatic effects. The significant non-climatic effect is then attributed to forest expansion. However, I have a few methodological concerns (detailed below) that leave me concerned about the robustness of the results. I

also find the analysis of the results to be fairly limited – the temperate vs. boreal differences are barely discussed for example, nor is the amount of variability in climatic and vegetation drivers within each biome (despite data on this clearly being used before aggregation in this study). The only real result presented is a qualitative statement of relative dominance that confirms previous studies.

My methodological concerns are as follows:

1) It is argued that forest inventory data cannot be used because they represent too large of an area (e.g. a county that may be larger than the watershed of study within it). In response, the authors aggregate the data even further, to cover an even larger area! How do we know that forest changes and climatic changes are consistent across all of the temperate and all of the boreal areas? The authors should assess the spatial variability of both forest inventory and rainfall data in each biome to ensure this is a reasonable approach

2) Similarly, LAI is calculated by using a constant leaf mass per area and biomass data from biome-aggregated NFI data (I think...the exact treatment of the NFI data is not clearly explained in Sec. 2.4). The authors then argue that LAI and areal forest cover is constant even as biomass increases by 23%. This would imply a huge trend in stem and branch biomass without any changes in other forest properties, which seems somewhat unlikely. Have the authors checked whether there are changes in forest composition over that time? What is the uncertainty induced in the LAI calculation based on assuming constant LMA for two species, and no other species contributions,however small? Furthermore, the statement that LAI is constant on page 6 line 32, directly contradicts the statement that LAI is changing on page 7 line 9.

3) Even for a catchment with unchanging vegetation conditions, there can be quite a lot of scatter on where a specific catchment's point falls relative to a theoretical Budyko curve due to interannual variability and imperfects in the Budyko framework. While a 50-year average may reduce noise to some degree, the entire climatic vs. non-climatic

calculation is potentially highly sensitive to the exact value of n used. Some bootstrapping and uncertainty propagation for n would be really helpful for demonstrating the results are robust.

4) As both the introduction and discussion mention, changes in the fraction of precipitation falling as snow could have a significant effect on the evaporative ratio (Berghuijs et al., 2014) that is not captured in the present analysis. A study of similar effects in China studying the effects of such a change is dismissed for making unrealistic assumptions, but that does not mean that the change itself could not be a factor here. The authors should at a minimum check if there are trends in the fraction of precipitation falling as snowfall. This is particularly troubling since Figure 5 shows a significant change in the seasonal cycle of rainfall in temperate areas.

There are several areas in which the presentation of this paper could be significantly improved

1) The specific Ep dataset used in Figures 3-6 is never stated.

2) I find Figure 4 quite hard to follow. Why are the colors not the same across the 4 sub-plots? This would be easier to read. If the colors represent the radius of each paddle, why are different paddles reaching the same radius colored different (e.g. 4a). Also, how is the r chosen for each paddle, given that it presumably represents multiple catchments?

3) Figure 6 suggests differences in the climatic vs non-climatic effects magnitudes between boreal and template. Possible reasons for these differences should be mentioned in the Discussion section, since this is one of the main ways in which your analysis allows detailed study. For example, are there differences in composition.

4) Can the authors comment on whether possible changes in air quality may play a role?

Other minor comments:

Page 2, line 40: Typo – formal?

Page 3, line 15: Would be helpful to explain 1986 is the midpoint of your data period

Page 7, line 16: This is not really a conflict with global studies. Even if global average trends are a certain way, showing that a specific location doesn't follow them is not a contradiction but indeed just a sign of spatial variability – CO2 effects can still dominate elsewhere and therefore for the global average cycle. However, see also Swann et al, PNAS 2016 for additional discussion on this topic.

Page 7, line 33: That "most of [drainage] was implemented before the present study period" conflicts with you statement that there is a peak in forest drainage implementation in the late 1970's and 80's (line 31)

Page 8, line 7-8: This sentence ("The fact that the upward . . .") is quite hard to follow.

---

## Referee Comment (RC4) · Anonymous Referee #4 · 19 Jul 2017

The manuscript by Jaramillo et al., 2017 analyses long term changes in ET/P and PET/P of Swedish catchments. The topic is of general interest and well suited for the journal.

Many catchments show increasing ET/P even though the aridity index is decreasing due to slightly higher precipitation rates. The data is compared with forest inventory data which shows a significant increase in forest biomass. The data suggests that the

overall increase in biomass is the dominant driver in increasing ET and thus ET/P. The authors argue that this "overrides physiological water saving responses". I do agree, however, the improved water use efficiency due to higher $CO_2$ levels might still be an important effect and could decrease ET, but clearly only for the same amount of biomass. For biomass aggregated results are presented, but there is no estimate of the physiological water saving response. Furthermore, the time series data does not provide statistical correlations between biomass and ET/P. Thus the study can not provide quantitative links between biomass or physiological water saving response and ET/P. However, both topics are suggested by the title and hypotheses. Therefore I recommend to adapt the red line of the manuscript or improve the analysis. Nevertheless, the observation that increases in forest biomass are potentially linked with increasing ET/P is important and should be communicated.

---

## Author Comment (AC1) · 8 Aug 2017

Response to Reviewer Nr. 1

We thank Reviewer Nr. 1 for highlighting the importance of our study and for proposing valuable suggestions to improve the manuscript. We have addressed below each of the Reviewers remarks, questions and suggestions.

Anonymous Referee #1,

Reviewer 1: Overview. This manuscript addresses causes of water balance changes in Swedish forests during the period 1961 to 2012. Water balance changes were encoded in estimates of the evaporative fraction, i.e., annual actual evapotranspiration over precipitation (E/P). The estimated changes in E/P were explained by climatic changes due to precipitation and potential evapotranspiration and by ecosystem changes due to standing forest biomass. The authors conclude that E/P increased 1961-2012 and attribute this increase to increased forest cover, despite a concurrent increase in precipitation. (i.e., decrease in the aridity index). The authors anticipated a reduction in E/P due to CO2 fertilization and interpret observed increased E/P as evidence that increased forest area overcompensated for any CO2 fertilization effect. The authors address a question of broad interest with relevance to water and carbon budgets from watershed to global scales. As noted by the authors, recent high profile papers have addressed the CO2 fertilization effect on the water balance directly (Betts et al. 2007) and in conjunction with climate, land use, and leaf area (Piao et al. 2007). As another example, Swann et al. (2016) and Milly and Dunne 2016) both suggest CO2 fertilization-induced decreases in ET may partially mitigate currently projected changes in continental drying and drought severity. The current paper follows in the footsteps of Piao et al. (2007) by addressing the effect of reforestation on basinscale evapotranspiration in the context of simultaneous CO2 and associated climatic changes. In general, the hypothesis and analysis were well thought out and executed.

Response 1: Thank you for this contextualization of our work and for appreciating its contributions.

Reviewer 2: Broadly, I think the clarity of the paper could be improved. And, more specifically, I have some difficulty understanding the analysis linking evapotranspiration changes to forest expansion rather than CO2 fertilization. "Possible physiological water saving response to rising CO2" The authors state in their title that the ET increase from forest expansion overrides ET decrease from rising CO2. The authors
then present evidence that (1) the aridity index decreased over time due to an increase in precipitation; (2) this decrease in aridity index lead to a decrease in E/P, as expected from the Budyko curve; and (3) there was an overall increase in E/P. This increase in E/P was then attributed to changes in forest standing biomass. There was no evidence or estimate of the CO2 fertilization effect and, therefore, I find it difficult to conclude that this effect was present and indeed over-compensated. To address this issue, I would suggest the authors reduce the focus on CO2 fertilization in the title and abstract.

Response 2: Thank you for this valuable remark. It is true that we do not provide direct evidence or quantification of an effect of a stomatal CO2 response on the evaporative index. What we do show is that such effect is either inexistent or weak, and considerable smaller than that of increasing forest biomass. We will reformulate both the title and abstract to acknowledge this. The overall conclusion of our study is not that there is a weak stomatal water-saving response to increasing CO2 but rather that increasing forest biomass due to forest management is an important driver of evapotranspiration change in this region. The reformulated title reads like this: "Increasing biomass drives evapotranspiration change in Swedish forests: Interpretation of movement in Budyko space"

Reviewer 3: My intuition is that the CO2 fertilization effect was relatively weak in this ecosystem. The atmospheric CO2 concentration increased approximately 85 ppm (315 too 400 ppm) over the study period, 1961-2012. In a meta-analysis of the FACE experiments (Ainsworth and Rogers 2007), trees showed one of the lowest responses of stomatal conductance to elevated CO2 (20% decrease). In these experiments, CO2 was increased from 366 ppm to 567 ppm, an increase 2.35 times that experienced from 1961 to 2012. As a first guess, one might expect less than 10% decrease in stomatal conductance whereas the authors show that forest standing biomass increased 25% and 55% in boreal and temperate watersheds (Figure 8). Further, in several places, the authors make the same argument based on results of species-specific responses. See page 2, lines 9-20 and page 7, lines 14-23. To paraphrase, the watersheds studied

are dominated by coniferous species and CO2 water saving response has not been observed in these species. In conclusion, there is little evidence to expect a CO2 water saving response in the studied watersheds. Therefore, I do not think it is appropriate to set up the paper with this hypothesis that is later not supported with the data. On the other hand, I do think it is appropriate to address this weak CO2 effect as one reason why an increase in E/P was observed, as the authors have done on page 7.

Response 3: We also thank the reviewer for this valuable comment and appreciate the calculations done to support it. We agree that many studies have shown that common gymnosperm species (e.g. conifers) in Northern Eurasian forests exhibit small stomatal responses to elevated CO2, as we also acknowledge on lines 15 to 20 of page 2. As mentioned in Response #2 above, we will remove the reference to CO2-induced stomatal water-savings from the title and reformulate the abstract to address this concern of the reviewer. Nevertheless, we think that the possible stomatal water-saving response to increasing CO2 should still be partially included in the abstract, introduction and main thread of the manuscript due to the following:

1. Although gymnosperms often exhibit small or inexistent reductions in stomatal conductance in response to elevated CO2, this is not always the case. For example, a Finnish field experiment showed that elevated CO2 decreased stomatal conductance in Scots pine (Kellomäki and Wang, 1996), as mentioned on lines 19 to 20 in Page 2. It is therefore possible a priori that conifers in our ecosystems exhibit significant CO2-induced stomatal closure responses. In addition, the most common deciduous tree species, Silver birch, exhibited marked elevated CO2-induced reductions in stomatal conductance in another field experiment (Rey and Jarvis, 1998), as mentioned in line 22 Page 7. Deciduous forest biomass accounted for up to 20% and 18% of total forest biomass in the temperate and boreal basin-groups, respectively, and their stomatal response to rising CO2 is thus potentially important for ET trends in our ecosystems.

2. Even though there are experimental indications that Northern conifers may exhibit weak responses of stomatal conductance to elevated CO2, vegetation models predict

equal CO2-induced reductions in stomatal conductance in conifers and gymnosperms. The reason for this is that the coupled stomatal–photosynthesis model used in vegetation and ecosystem models (Ball et al., 1987; Leuning, 1995; Medlyn et al., 2011) have a general formulation of the stomatal CO2 response for all species. It is therefore highly relevant for our study to assess the likelihood for substantial CO2-induced stomatal water-saving responses for our region (as for example predicted by Luo et al. (2008) or Betts et al. (2007), or if this response is much less important for trends in evapotranspiration than changes in the landscape (e.g. increases in forest biomass).

3. The analysis of long-term monitoring data in the present study tests if responses observed in manipulative field experiments and predicted by ecosystem models agree with field observations in Swedish forests over the past 50 years. Field experiments suffer from being conducted at limited temporal and spatial scales; most tree experiments investigating effects of rising CO2 on stomatal conductance and transpiration in Northern forest trees are only a couple of years long and include a low number of trees. Models heavily rely on knowledge of mechanisms and processes gained in experiments and thus gain from being thoroughly evaluated for ecological realism using monitoring data. We therefore think that our methodology is a valuable tool for evaluating large-scale and long-term applicability of both experimental responses and modelling predictions. By showing that changes in forest biomass are more important than possible (if existing) stomatal closure responses to rising atmospheric CO2 in controlling ET/P change over the past 50 years, our study highlights the importance for models to accurately capture changes in the landscape when projecting hydroclimatic change of this region. In addition, it shows that predictions of large-scale CO2-induced stomatal water-saving responses are unlikely to hold for our type of ecosystem. We will clarify these three points in the Introduction and Discussion of a revised manuscript. We will also add in the Introduction the additional references suggested by the reviewer (Milly and Dunne, 2016; Swann et al., 2016) and one additional study (Prudhomme et al., 2014) to highlight the importance of understanding the existence of the water saving response in Northern latitudes due to its implications for future estimates of evapotranspiration and drought.

Reviewer 4: Specific comments Hydroclimatic Data: 1) The temporal scales of the data are inconsistent. For the Penman-Monteith model, the long-term mean annual geostrophic wind at 1000 meters above sea level is used. Given fine-scale variability in windspeed and its local, leaf-scale effect on transpiration, this approach is not warranted. This is especially true for comparison with the Langbein and Hargreaves models, which use daily temperature as model input. My suggestion is to remove the Penman-Monteith analysis and use the 3 other sources for PET.

Response 4: Thanks for this suggestion. We will remove the Penman-Monteith estimate of potential evapotranspiration from the analysis and methodology, still leaving the other remaining three methods of calculation.

Reviewer 5: 2) The discussion in the first paragraph surrounding equation (1) is disorganized. My suggestion is to place the description of the P data before equation (1). Then follow with the sentence, "We used annual P and R data to calculate : : :" after you've described what the annual P and R data are.

Response 5: Thanks for the suggestion. We will do as suggested by the Reviewer. Budyko Framework:

Reviewer 6: 1) I am not familiar with the Psi notation for evaporative index and typically see it written as E/P or something similar. It may help the reader to be consistent with notation from your references.

Response 6: We will change the notation as suggested by the Reviewer, so that notation is consistent with former references.

Reviewer 7: Linking the residual effect Delta Psi_r to forest change: 1) I don't understand equation 8. It is described as a five-year moving average, but there is only a j and j+1 term – does this mean only the current and previous year are used in the calculation?

Response 7: Sorry for the misunderstanding. The 5-year moving window was applied after (not before) the resulting annual time series of the residual effect on the evaporative ratio is obtained from Eq. 8. We added this five-year window to the annual results of the residual of the evaporative ratio in Figure 8 to help visualizing the evaporative ratio trends. Furthermore, we have noticed that Eq. 8 was misleading and confusing, as noticed also by Reviewer 2. The "cumulative of the residual change of the evaporative ratio [$\Delta\Psi r$]" was definitely an unnecessarily complicated way to express such variable. It should have been described as "the residual ($\Psi r$) obtained when subtracting for each year the climatic estimate of the evaporative ratio $\Psi c$ (Eq. 4) from the estimate calculated through the water budget equation $\Psi$ (Eq. 1 and 2)", i.e. $\Psi r=\Psi-\Psi c$. In a revised manuscript, we will remove Eq. 8, clarify this instead and do the corresponding changes in the caption of Figure 8. We will also remove the use of the 5-year moving window. We have here performed a statistical analysis on the annual data and show that our results are still robust regardless of the use of this 5-year moving window. Please see Response 4 for to Reviewer 2.

Reviewer 8: Figure 2, Can you re-orient the arrow from t1 to t2 to be consistent with your result of decreasing aridity index? That would make the figure easier to read.

Response 8: We have accordingly modified Figure 2, please see below. We will include it as such in a revised manuscript.

Reviewer 9. Figure 3: Can you include a separate Budyko plot for the early and late periods? That would give the reader a general, more intuitive sense of how the watersheds moved in the Budyko space.

Response 9: We will include these figures in the Supplementary information. Please see them below. We thank the Reviewer for this suggestion since with these figures show why movements in Budyko space for a large set of basins are difficult to differentiate without the use of wind-rose type plots of Fig. 4.

Reviewer 10, Figure 6: What is the y-axis label in this figure? Same for Figure 7.

Response 10: The y-axis is the change in any of the evaporative ratio index and its components, described under each box plot distribution. We will add this accordingly.

References

Ball, J. T., Woodrow, I. E. and Berry, J. A.: model predicting stomatal conductance and its contribution to the control of photosynthesis under different environmental conditions, Prog. Photosynth. Res. Proc. VIIth Int. Congr. Photosynth. Provid. R. I. USA August 10-15 1986 Ed. J Biggins [online] Available from: http://agris.fao.org/agris-search/search.do?recordID=US201301403137 (Accessed 7 August 2017), 1987.

Betts, R. A., Boucher, O., Collins, M., Cox, P. M., Falloon, P. D., Gedney, N., Hemming, D. L., Huntingford, C., Jones, C. D., Sexton, D. M. H. and Webb, M. J.: Projected increase in continental runoff due to plant responses to increasing carbon dioxide, Nature, 448(7157), 1037–1041, doi:10.1038/nature06045, 2007.

Kellomäki, S. and Wang, K.-Y.: Photosynthetic responses to needle water potentials in Scots pine after a four-year exposure to elevated $CO_2$ and temperature, Tree Physiol., 16(9), 765–772, doi:10.1093/treephys/16.9.765, 1996.

Leuning, R.: A critical appraisal of a combined stomatal‐photosynthesis model for C3 plants, Plant Cell Environ., 18(4), 339–355, doi:10.1111/j.1365-3040.1995.tb00370.x, 1995.

Luo, Y., Gerten, D., Le Maire, G., Parton, W. J., Weng, E., Zhou, X., Keough, C., Beier, C., Ciais, P., Cramer, W., Dukes, J. S., Emmett, B., Hanson, P. J., Knapp, A., Linder, S., Nepstad, D. and Rustad, L.: Modeled interactive effects of precipitation, temperature, and [CO2] on ecosystem carbon and water dynamics in different climatic zones, Glob. Change Biol., 14(9), 1986–1999, doi:10.1111/j.1365-2486.2008.01629.x, 2008.

Medlyn, B. E., Duursma, R. A., Eamus, D., Ellsworth, D. S., Prentice, I. C., Barton, C. V. M., Crous, K. Y., De Angelis, P., Freeman, M. and Wingate, L.: Reconciling the optimal and empirical approaches to modelling stomatal conductance, Glob. Change

Biol., 17(6), 2134–2144, doi:10.1111/j.1365-2486.2010.02375.x, 2011.

Milly, P. C. D. and Dunne, K. A.: Potential evapotranspiration and continental drying, Nat. Clim. Change, 6(10), 946–949, doi:10.1038/nclimate3046, 2016.

Prudhomme, C., Giuntoli, I., Robinson, E. L., Clark, D. B., Arnell, N. W., Dankers, R., Fekete, B. M., Franssen, W., Gerten, D., Gosling, S. N., Hagemann, S., Hannah, D. M., Kim, H., Masaki, Y., Satoh, Y., Stacke, T., Wada, Y. and Wisser, D.: Hydrological droughts in the 21st century, hotspots and uncertainties from a global multimodel ensemble experiment, Proc. Natl. Acad. Sci., 111(9), 3262–3267, doi:10.1073/pnas.1222473110, 2014.

Rey, A. and Jarvis, P. G.: Long-term photosynthetic acclimation to increased atmospheric CO2 concentration in young birch (Betula pendula) trees, Tree Physiol., 18(7), 441–450, doi:10.1093/treephys/18.7.441, 1998.

Swann, A. L. S., Hoffman, F. M., Koven, C. D. and Randerson, J. T.: Plant responses to increasing CO2 reduce estimates of climate impacts on drought severity, Proc. Natl. Acad. Sci., 113(36), 10019–10024, doi:10.1073/pnas.1604581113, 2016.
* * *
[Figure]

**(a)**

[Figure]

**(b)** Expanding forest

Water saving response to increasing $CO_2$

$(E/P)_c = (1 + (E_p/P)_1^{-n})^{-1/n}$

**Fig. 1.**

[Figure]

Supplementary Figure for Budyko space- both periods

[Figure]

[Figure]

**Fig. 2.**

---

## Author Comment (AC2) · 9 Aug 2017

Response to Reviewer Nr. 2

We thank Reviewer Nr. 2 for highlighting the importance of our study, for appreciating our appreciating our "convincing arguments" and for proposing valuable suggestions to improve the manuscript. We have addressed below each of the Reviewers remarks, questions and suggestions.

[Figure]

Reviewer 1: This manuscript uses climatic (temperature and precipitation), vegetation (forest expansion in Sweden), and runoff time series from 65 unregulated Swedish basins over 1961-2012 to investigate changes in the precipitation partitioning into evapotranspiration (ET) and runoff. The authors are specifically interested in seeing if increase in forest biomass that occurred in the past decades would combine with two competing physiological phenomena to either increase or decrease ET beyond the extent dictated by climate (represented by the aridity index): (1) decrease plant stomatal conductance in response to increase in $CO_2$ (water saving responses), resulting in a decrease in ET, or (2) $CO_2$-induced increase in plant growth and leaf area, resulting in a increase in ET. The contribution of this manuscript is thus organized into two main components: that of analysis of change, and of attribution of this change to forest properties (total area, volume, and proportion of deciduous species to total LAI). In my opinion, the authors have made a convincing argument for residual changes in basin-level ET that goes the extent dictated by climate. They have postulated that, because the observed ET has increased despite a decrease in aridity index (when Budyko's curve, under stationary conditions, would suggest otherwise based solely on climatic effects), there must exist some non-climate related mechanisms that offset this increase.

Response 1: We thank the Reviewer for the good summary of our work and for appreciating our "convincing arguments" and analyses.

Reviewer 2: To make this point, however, I think that Figure 4 is redundant with Figure 6. Figure 4's use of "wind roses" does not add additional support for the authors' main point. While they claim that these wind roses are "a simple way to synthetize general tendencies of movement," the general direction of these movements are well summarized by the histograms presented in Figure 6, so to me these are two different graphical representation for very similar sets of information. In addition, "spectra of movements in Budyko space," used repeatedly in Section 3.1, need to be rephrased. Since "spectra" has a very specific meaning in time series analysis, I would suggest

the authors avoid this term in reference to movement in the Budyko coordinates.

Response 2: Thank you for both suggestions. First, we will remove any reference to "spectra" and refer to the wind-rose type diagrams as "roses" instead. We hope that this change will satisfy the reviewer. The reviewer is right in the fact that the roses (Fig. 4) and the boxplots (Fig. 6) show similar data. However, we still consider both are important for the red line of the manuscript and we think that the problem was the fact that we did not justify enough the simultaneous use of both figures. We will expand on the usefulness of each figure when analyzing the results of each figure in a revised manuscript, as follows:

In summary, the roses (Figure 4) show the direction and magnitude of movement in Budyko space from the first period to the second period for each basin as typical wind roses show wind direction and wind speed. As such, this representation enables the simultaneous study of changes in both evaporative ratio and aridity index and their distribution among individual basins. This cannot be done with the boxplots of Figure 6. By following the analysis work of Jaramillo and Destouni (2014), these roses give an opportunity to estimate the percentage of basins in each basin group for which a change in potential evapotranspiration and precipitation is not enough to explain the observed change in the evaporative ratio (approximately 60% in the case of both our basin groups). We consider this an important result, and have included it in the abstract (Line 23-25 Page 1) that could not be obtained from the boxplots of Figure 6. The roses also allow help synthetize movement in Budyko space for large number of basins since such movements are small and difficult to identify at the complete scale of the Budyko space plot (see Reponse 9 to Reviewer 1).

On the other hand, the box plots of Figure 6 present the characteristics of each basin-group distribution (quantiles, outliers, and median) of the total, residual, and climatic components of the evaporative ratio as well as their arithmetic and area-weighted means for each group. This would be a difficult task with the roses. To address the reviewers concern, we have also used Figure 6 to include an uncertainty analysis (Please

see figure below and Response 2 of Reviewer 3 for more information on the uncertainty analysis) requested by reviewer 3, differentiating it even more from Figure 4. We hope that the Reviewer considers that this explanation supports the use of both figures.

Reviewer 3: I think also that the weakness of the manuscript as it stands lies in formulating the argument for the second point, e.g., in attributing the observed increase in ET to a specific, hypothesized mechanism. In Figure 8, boreal and temperature forests showed opposite changes in this deciduous proportion, though how this might contribute to the overall increase in ET in both forest types is not discussed.

Response 3: Thanks for this valuable suggestion. Our intention was not to attribute the increase in the residual of the evaporative ratio to a single mechanism but rather to assess which forest attribute may best explain that increase. We will revise the title and abstract of the manuscript accordingly. We will reaffirm in a revised manuscript that the resulting change in the residual component of the evaporative ratio is related to changes in "forest structure", and that in turn the forest structure attribute that best explained the observed changes in the residual component of change in the evaporative ratio was forest biomass, followed by forest cover.

Changes in the composition of the forest (another forest structure attribute) may also have an effect in the evaporative ratio of these basin groups, however, our results showed no statistically significant change in forest composition (i.e., the ratio of deciduous leaf area index to total leaf area index, called originally LAIQ and from now on QLAI) in either of the two basin groups (Fig. 8 and 9). However, we will specify this in a clearer way, to say that the uncertainty of the experiment did not allow the detection of changes in forest composition. We also found a small mistake in the calculation of forest composition change in Figure 8 that was not showing this result in a clear way, so we have updated Figure 8 accordingly. See updated Figure below.

Reviewer 4: In addition, the relationship between forest attributes and ïĄĎΨr is described in Section 3.2 using very vague terms like "in agreement with" and "followed

that of." I would suggest applying more statistical analysis (and plot out the correlation between ïĄĎΨr and each of the forest attributes) in this section to more quantitatively describe these relationships.

Response 4: Thanks for the suggestion. A statistical analysis would greatly improve the robustness of the conclusions of the manuscript. To further address the concern of the Reviewer, we have added the calculation of the coefficient of determination ($R^2$) of the linear regression between all obtained annual values of the residual component of the evaporative ratio ($\Psi r = \Psi - \Psi c$) and the annual values of the three mentioned attributes of forest structure for the temperate and boreal basin groups. We found that forest biomass (V) explain most of the variance of $\Psi r$; the $R^2$ for the relationship between forest biomass and $\Psi r$ is significantly different from zero ($p < 0.05$) for both the boreal and temperate groups. In turn, $R^2$ for the relationship between forest cover and $\Psi r$ is only significantly different from zero ($p < 0.05$) in the temperate group. Forest composition does not have any significant relationship with $\Psi r$ in any of the two basin groups. We will present these results as Table 1 in a revised manuscript (see below).

Reviewer 5: I also remain unconvinced of the authors' use of the cumulative [$\Psi r$] in comparison to the forest attributes (Figure 9), and the application and choice of a 5-year moving window for [$\Psi r$]. Both of these usages require further justification.

Response 5: Thanks for expressing these concerns. As expressed in our Response 7 to Reviewer Nr. 1, we are sorry to say that the expression "the cumulative of the residual change of the evaporative ratio [$\Delta \Psi r$]" was definitely an unnecessarily complicated way to define such variable. We had decided to define it as such in order to continue with the notation of change. However, it should have been described as "the residual $\Psi r$ obtained when subtracting for each year the climatic estimate of the evaporative ratio $\Psi c$ (Eq. 4) from the estimate based calculated from the water budget equation $\Psi$ (Eq. 1 and 2)", i.e. $\Psi r = \Psi - \Psi c$. In a revised manuscript, we will remove Eq. 8, clarify this instead and do the corresponding changes in the caption of Figure 8.

[Figure]

We will also remove any use of the five-year moving window in a new manuscript since the new statistical analysis suggested by the reviewer and now applied on the annual results of $\Psi$r (without the moving window) our robust enough (See Response 4 to this Reviewer).

Reviewer 6: If the authors can address these concerns, this paper will make a good contribution to the study of water partitioning at high latitudes.

Response 6: Thank you!

References

Jaramillo, F. and Destouni, G.: Developing water change spectra and distinguishing change drivers worldwide, Geophys. Res. Lett., 41(23), 8377–8386, doi:10.1002/2014GL061848, 2014.

Figure 6-Modified

[Figure]

**Fig. 1.**

Corrected Figure 8

[Figure]

**Fig. 2.**

New Table 1-

**BOREAL**

| | biomass | forest cover | forest composition | fraction falling as snow |
|---|---|---|---|---|
| | $V$ | $A$ | $Q_{LAI}$ | $f_s$ |
| intercept | -0.09 | 0.44 | -0.10 | 0.11 |
| slope | 0.00 | -0.78 | 0.89 | -0.15 |
| Adjusted $R^2$ | 0.07 | 0.02 | -0.02 | 0.02 |
| p | 0.028* | 0.168 | 0.629 | 0.187 |

**TEMPERATE**

| | biomass | forest cover | forest composition | fraction falling as snow |
|---|---|---|---|---|
| | $V$ | $A$ | $Q_{LAI}$ | $f_s$ |
| intercept | -0.06 | -0.25 | -0.16 | 0.02 |
| slope | 0.00 | 0.41 | 1.09 | -0.05 |
| Adjusted $R^2$ | 0.08 | 0.06 | 0.03 | -0.01 |
| p | 0.026* | 0.048* | 0.168 | 0.612 |

**Fig. 3.**

---

## Author Comment (AC3) · 9 Aug 2017

Response to Reviewer Nr. 3

We thank Reviewer Nr. 3 for explaining in detail the concerns regarding the robustness of our results as well as the suggestions proposed to increase the robustness of the analysis. We have now performed an uncertainty analysis and a statistical assessment that improve the robustness of our results. We have also addressed below each of the

[Figure]

Reviewers remarks, questions and suggestions.

Reviewer 1: This paper uses the Budyko framework to study the effect of changes in evaporative ratios at a number of boreal and temperate catchments in Sweden. The study looks at changes in the location of each catchment in Budyko space during two consecutive 25-year periods in the early 21st century and second half of he 20th century, and separates the changes into climatic and non-climatic effects. The significant non-climatic effect is then attributed to forest expansion. However, I have a few methodological concerns (detailed below) that leave me concerned about the robustness of the results. I also find the analysis of the results to be fairly limited – the temperate vs. boreal differences are barely discussed for example.

Response 1: We thank the reviewer for the recommendations. We address each of these concerns below. Regarding the differences between the boreal and temperate basin groups, we will take care to discuss in more detail the similarities and differences in responses in a revised manuscript.

We will also expand on the difference in the species composition between both groups, in terms of the mean biomass of each species in the initial period 1961-1986. We will include this as a Panel c in Figure 1 and discuss the main differences accordingly (oranges are deciduous species and blues are coniferous species). The corresponding change is then the one originally shown in Figure 8.

Reviewer 2: The only real result presented is a qualitative statement of relative dominance that confirms previous studies nor is the amount of variability in climatic and vegetation drivers within each biome (despite data on this clearly being used before aggregation in this study). My methodological concerns are as follows: 1) It is argued that forest inventory data cannot be used because they represent too large of an area (e.g. a county that may be larger than the watershed of study within it). In response, the authors aggregate the data even further, to cover an even larger area! How do we know that forest changes and climatic changes are consistent across all of the temperate and all of the boreal areas? The authors should assess the spatial variability of both forest inventory and rainfall data in each biome to ensure this is a reasonable approach.

Response 2: We thank the Reviewer for bringing this important issue that although inherent of the Swedish Forest Inventory in itself, must be clarified here. Let us answer the Reviewer here by first doing some clarifications on the forest data to show that we have already dealt with spatial uncertainty.

Forest data

The idea here is not to describe again the entire methodology which is available in Fridman et al. (2014), but rather to clarify the necessary aspects required to answer the Reviewer's concern.

The data of the Swedish National Forest Inventory (NFI) utilizes a stratified systematic sample based upon clustered sample plots and designed to deliver statistics at county level, mainly of forestland. The NFI was first undertaken in 1923 in the form of a belt inventory, and since 1953 has become a systematic cluster sample inventory. Since 1983, the sampling scheme also includes permanent sample plots, providing a greater precision in change estimates of forest characteristics. The NFI already accounts for changes in methodology across time. The main forest attributes used here to study forest development in Sweden are the area, standing volume and leave/needle volume of productive forest differentiated into several categories (i.e., species, diameter and age composition, forest management stage). The strata of the NFI are based on the Swedish counties; the sample plots have been distributed within each stratum. A single sample distribution is completed every five years, however as each year (representing a fifth of the sample) is evenly distributed over the country, any consecutive five year period can be used.

The sample plots are not restricted to provide only county wise estimates – every sample plot has an upscaling factor and therefore by using a GIS-layer polygon any group

of sample plots can be used to create estimates of forest attributes for such polygon. However, a larger polygon will result in more sample plots being used and a lower standard error. Many of the original 65 basins contained too few sample plots to provide meaningful estimates and therefore the basins were aggregated into larger groups. This is the reason why we decided to aggregate the 65 basins into two main basin groups in order to show the change in forest statistics within the 65 basins and reduce the sampling standard error. The sampling standard error of such calculation and spatial aggregation is already shown and quantified in Figure 8 and mentioned in its caption for both area and volume data. The sampling error for the LAIQ (from now on QLAI) estimate is the propagation of the corresponding sampling errors of forest area and volume statistics (See caption of the Figure).

We acknowledge that there will be a spatial variability within the boreal and temperate basins that is not accounted for. We originally tested with smaller basin groups but the resulting standard errors were too large. Although there is a difficult in achieving the best balance between spatial resolution and the statistical uncertainty associated with a sample based inventory, we think that the selected basin-groups used for the analysis represent the best approach to address them.

Precipitation

Now, regarding the spatial uncertainty of precipitation (P), as mentioned in the manuscript, we have used a widely used methodology, the Thiessen polygon method, to interpolate spatially the spatial data of the 68 precipitation stations with best availability that are located within and near the 65 basins. However, in order to address the reviewers concern, we will use two more precipitation products to calculate mean annual precipitation values for each basin, and incorporate them in an uncertainty assessments. These products are the precipitation estimate from the Climatic Research Unit-gridded P product CRU TS3.23 (Harris et al., 2014) and the mean daily P product of the Luftwebb (http://luftwebb.smhi.se/) portal of the Swedish Meteorological and Hydrological Institute (SMHI). The latter is a gridded dataset of precipitation for Sweden during the period 1961-2014 with a 4 x 4 km horizontal resolution that is based in data collected from over 87 precipitation stations around the country (Johansson, 2000; Johansson and Chen, 2003).

We have now combined the three precipitation P products with the three potential evapotranspiration (E0) products mentioned in Page 3 lines 20 and 28 (excluding the Penman-Monteith estimate after following Reviewer Nr. 1 suggestion), to obtain a total of nine possible combinations of P and E0. As such, we now have three estimates of change in the evaporative ratio ($\Delta\Psi$) and nine of each of its components ($\Delta\Psi c$ and $\Delta\Psi r$). We have generated a corresponding uncertainty range (Modified Figure 6, see below) for the calculations of the arithmetic average (blue vertical range of uncertainty) and area-weighted average (red vertical range of uncertainty) for $\Delta\Psi$, $\Delta\Psi c$ and $\Delta\Psi r$ in each of the two basin groups; temperate and boreal. We found that regardless of the combination of P and E0, $\Delta\Psi c$ is always negative and $\Delta\Psi r$ is always positive, supporting our previous results and evidencing their robustness. We thank the Reviewer for suggesting this analysis. We will add this new methodology, analysis and figure in a revised manuscript.

Reviewer 3: 2) Similarly, LAI is calculated by using a constant leaf mass per area and biomass data from biome-aggregated NFI data (I think the exact treatment of the NFI data is not clearly explained in Sec. 2.4). The authors then argue that LAI and areal forest cover is constant even as biomass increases by 23%. This would imply a huge trend in stem and branch biomass without any changes in other forest properties, which seems somewhat unlikely. Have the authors checked whether there are changes in forest composition over that time? What is the uncertainty induced in the LAI calculation based on assuming constant LMA for two species, and no other species contributions, however small? Furthermore, the statement that LAI is constant on page 6 line 32, directly contradicts the statement that LAI is changing on page 7 line 9.

Response 3: We must clarify that our results do not show that the leaf area index of deciduous and temperate forest cover is constant in time. What is constant in time is the

ratio of deciduous LAI to total LAI (LAIQ), used here as a proxy for forest composition (See section 2.4). We will change the name of this variable to QLAI throughout a revised manuscript to avoid this confusion, and reword on page 6 line 32. Changes in the composition of the forest (another forest structure attribute) may also have an effect in the evaporative ratio of these basin groups, however, our results showed no statistically significant change in forest composition due to a large propagated sampling standard error (Figs. 8 (new updated version below) and Fig. 9). However, we will specify this in a clearer way, to say that the uncertainty of the experiment did not allow the detection of changes in forest composition.

Reviewer 4: 3) Even for a catchment with unchanging vegetation conditions, there can be quite a lot of scatter on where a specific catchment's point falls relative to a theoretical Budyko curve due to interannual variability and imperfects in the Budyko framework. While a 50-year average may reduce noise to some degree, the entire climatic vs. non-climatic calculation is potentially highly sensitive to the exact value of n used. Some bootstrapping and uncertainty propagation for n would be helpful for demonstrating that the results are robust.

Response 4: Thanks for this reminder. The uncertainty of n in Budyko type equations should be taken into account in order to obtain a proper range of uncertainty of the annual climatic estimate of the evaporative ratio (Greve et al., 2015) and its corresponding changes in the components ïĄĎΨc and ïĄĎΨr. Based on the nine possible outcomes of $\Delta\Psi$r that are now available after following the uncertainty analysis described in Response 2 to this reviewer, we now have also nine possible n values for each basin and many more for each basin group. This, because we use the mean estimates of P and E0 to calculate an n value for each basin. The uncertainty ranges of ïĄĎΨ,ïĂăïĄĎΨc, ïĄĎΨr shown in the modified Figure 6 of Response 2 include the minimum and maximum area-weighted and arithmetic means obtained from propagating such uncertainties into the calculation of these components. The results show again the robustness of our results regarding the estimates of P, E0 and n. Again, these results prove the robustness of our results and we thank the reviewer for addressing this concern.

Reviewer 5: 4) As both the introduction and discussion mention, changes in the fraction of precipitation falling as snow could have a significant effect on the evaporative ratio (Berghuijs et al., 2014) that is not captured in the present analysis. A study of similar effects in China studying the effects of such a change is dismissed for making unrealistic assumptions, but that does not mean that the change itself could not be a factor here. The authors should at a minimum check if there are trends in the fraction of precipitation falling as snowfall. This is particularly troubling since Figure 5 shows a significant change in the seasonal cycle of rainfall in temperate areas.

Response 5: We agree on the importance of accounting for changes in the fraction of precipitation falling as snow (fs) (Berghuijs et al., 2014). That is why we had calculated these changes from the period 1961-1986 to the period 1987-2012 (Lines 17-18 Page 8 "Our calculations show that the fraction of precipitation falling a snow decreased from the period 1961-1986 to the period 1987-2012 from 0.20 to 0.14 in temperate basins and from 0.45 to 0.43 in boreal basins". Mean annual fs was calculated for each basin based on the collected daily P and T data; we assumed that precipitation in days with mean temperatures below 1 Íę C falls as snow and above 1 Íę C as rain, following Berghuijs et al. (2014).

To further address the concern of the reviewer, we have now performed an additional statistical analysis that calculates the coefficient of determination ($R^2$) of the linear regression between all annual values of $\Psi r = \Psi - \Psi c$ and fs in the two basin groups. We found that forest biomass explains more variance of $\Psi r$ than fs in both biomes. We found that $R^2$ for the linear regression between fs and $\Psi r$ is not significantly different from zero ($p > 0.05$) for either the boreal or the temperate boreal group. In more detail, fs can explain more of the variance of $\Psi r$ in boreal basins than in temperate basins. One explanation for this may be that the possible decrease in spring snow in boreal basins is associated to the observed increase in the length of the growing season due
to increasing temperatures (Hasper et al., 2016). A longer growing season may result in more annual transpiration, under constant annual precipitation conditions. These results also make our previous findings more robust and thank this the Reviewer for this other suggestion idea. We will include this new analysis, table and discussion in a revised manuscript.

Reviewer 6: There are several areas in which the presentation of this paper could be significantly improved 1) The specific Ep dataset used in Figures 3-6 is never stated.

Response 6: The data sets of P, T, Tmin and Tmax used to calculate E0 and the four models/products used for the calculation were mentioned and explained in Lines 20 to 29 of Page 3. The Penman-Monteith model will be removed from a revised manuscript to follow Reviewer Nr. 1's suggestion.

Reviewer 7: 2) I find Figure 4 quite hard to follow. Why are the colors not the same across the 4 sub-plots? This would be easier to read. If the colors represent the radius of each paddle, why are different paddles reaching the same radius colored different (e.g. 4a). Also, how is the r chosen for each paddle, given that it presumably represents multiple catchments?

Reponse 7: The roses (Figure 4) show the direction and magnitude of movement in Budyko space from the first period to the second period for each basin, in the same way that a typical wind rose shows wind direction and wind speed. The colored roses (green for boreal and purple for temperate basins) show movements in Budyko space as calculated from observations (where ET=P-R; Eqs. 1 and 2). The grey roses show instead movements due to climate only as calculated with the Chodbhury equation (Eq. 4). Each paddle in a rose groups all movements occurring in a range of directions in Budyko space ($\theta$) of 15 degrees. This value was chosen arbitrarily, to provide the sufficient detail of directions. As example (see below), 37% of all boreal basins (green rose) have moved in the range of directions ($270°<\theta<295°$, $\theta$ starts from the upper vertical and clockwise). Now, the intensity of the color describes the range of magnitudes

(r: dimensionless) of those movements in Budyko space. As example, of those boreal basins moving in the range of directions previously described (37% of the boreal basins), 14% have moved with magnitudes between 0 and 0.05 (light green), 14% with magnitudes between 0,05 and 0,10 (medium green) and 9% with magnitudes between 0.10 and 0.30 (dark green). On the contrary, no basin has moved in this range of directions ($270° < \theta < 295°$) when using the Chodbhury Equation (grey rose). We hope this additional explanation makes the use of the roses easier to understand. We will include this example explanation in the text of a revised manuscript for better understanding of these figures and improve the caption of the figure.

Reviewer 8: 3) Figure 6 suggests differences in the climatic vs non-climatic effects magnitudes between boreal and template. Possible reasons for these differences should be mentioned in the Discussion section, since this is one of the main ways in which your analysis allows detailed study. For example, are there differences in composition.

Response 8: Thanks. As mentioned in Response 1, we will expand on the discussion about the possible differences between the boreal and temperate basin groups. However, even after rechecking all our forest data, our results show no statistically significant change in forest composition (i.e., the ratio of deciduous leaf area index to total leaf area index, called now QLAI) in either of the two basin groups (Fig. 8 and 9).

Reviewer 9: 4) Can the authors comment on whether possible changes in air quality may play a role?

Response 9: Yes, since air pollution such as ground-level ozone and acid rain affects the canopy of the forest and the LAI, evapotranspiration and the evaporative ratio may be lower than under unaffected conditions (i.e., a negative change). Such is the case of the study by Renner et al. (2013) mentioned in Line 33 of Page 2. However, since the residual of change in the evaporative ratio is in our study positive, we can assume that the effect of air quality in this case is not as important as that of increasing biomass.

Reviewer 10: Other minor comments: Page 2, line 40: Typo – formal?

Response 10: We will remove the word "former"

Reviewer 11: Page 3, line 15: Would be helpful to explain 1986 is the midpoint of your data period

Response 11: We will mention that the two 26-year sub-periods are equal in length.

Reviewer 12: Page 7, line 16: This is not really a conflict with global studies. Even if global average trends are a certain way, showing that a specific location doesn't follow them is not a contradiction but indeed just a sign of spatial variability – CO2 effects can still dominate elsewhere and therefore for the global average cycle. However, see also Swann et al, PNAS 2016 for additional discussion on this topic.

Response 12: Thanks. We will include the reference mentioned by the reviewer and remove the mention of contradiction with other global studies.

Reviewer 13: Page 7, line 33: That "most of [drainage] was implemented before the present study period" conflicts with you statement that there is a peak in forest drainage implementation in the late 1970's and 80's (line 31)

Response 13: Thanks. We will rewrite to avoid this contradiction. Most of the drainage occurred before our study period, but there was also a later smaller peak within our period, in the late 1970's and 80's.

Reviewer 14: Page 8, line 7-8: This sentence ("The fact that the upward: :") is quite hard to follow.

Response 14: We will reword this sentence for more comprehension. Thanks again.

References

Berghuijs, W. R., Woods, R. A. and Hrachowitz, M.: A precipitation shift from snow towards rain leads to a decrease in streamflow, Nat. Clim. Change, 4(7), 583–586,

[Figure]

doi:10.1038/nclimate2246, 2014. Fridman, J., Holm, S., Nilsson, M., Nilsson, P., Ringvall, A. and Ståhl, G.: Adapting National Forest Inventories to changing requirements – the case of the Swedish National Forest Inventory at the turn of the 20th century, Silva Fenn., 48(3), doi:10.14214/sf.1095, 2014. Greve, P., Gudmundsson, L., Orlowsky, B. and Seneviratne, S. I.: Introducing a probabilistic Budyko framework, Geophys. Res. Lett., 42(7), 2015GL063449, doi:10.1002/2015GL063449, 2015. Hasper, T. B., Wallin, G., Lamba, S., Hall, M., Jaramillo, F., Laudon, H., Linder, S., Medhurst, J. L., Räntfors, M., Sigurdsson, B. D. and Uddling, J.: Water use by Swedish boreal forests in a changing climate, Funct. Ecol., 30(5), 690–699, doi:10.1111/1365-2435.12546, 2016. Johansson, B.: Areal Precipitation and Temperature in the Swedish Mountains, Hydrol. Res., 31(3), 207–228, 2000. Johansson, B. and Chen, D.: The influence of wind and topography on precipitation distribution in Sweden: statistical analysis and modelling, Int. J. Climatol., 23(12), 1523–1535, doi:10.1002/joc.951, 2003.

New Figure 1c

[Figure]

Fig. 1.

[Figure]

[Figure]

Figure 6-Modified

**Fig. 2.**

Corrected Figure 8

[Figure]

**Fig. 3.**

New Table 1-

**BOREAL**

|  | biomass | forest cover | forest composition | fraction falling as snow |
|---|---|---|---|---|
|  | $V$ | $A$ | $Q_{LAI}$ | $f_s$ |
| intercept | -0.09 | 0.44 | -0.10 | 0.11 |
| slope | 0.00 | -0.78 | 0.89 | -0.15 |
| Adjusted $R^2$ | 0.07 | 0.02 | -0.02 | 0.02 |
| p-value | 0.028* | 0.168 | 0.629 | 0.187 |

**TEMPERATE**

|  | biomass | forest cover | forest composition | fraction falling as snow |
|---|---|---|---|---|
|  | $V$ | $A$ | $Q_{LAI}$ | $f_s$ |
| intercept | -0.06 | -0.25 | -0.16 | 0.02 |
| slope | 0.00 | 0.41 | 1.09 | -0.05 |
| Adjusted $R^2$ | 0.08 | 0.06 | 0.03 | -0.01 |
| p-value | 0.026* | 0.048* | 0.168 | 0.612 |

**Fig. 4.**

---

## Author Comment (AC4) · 9 Aug 2017

Response to Reviewer Nr.4

We thank Reviewer Nr. 4 for highlighting that our results should be communicated and for proposing valuable suggestions to improve the manuscript. We have addressed below the Reviewers remarks, questions and suggestions.

[Figure]

Anonymous Referee #4

The manuscript by Jaramillo et al., 2017 analyses long term changes in ET/P and PET/P of Swedish catchments. The topic is of general interest and well suited for the journal. Many catchments show increasing ET/P even though the aridity index is decreasing due to slightly higher precipitation rates. The data is compared with forest inventory data which shows a significant increase in forest biomass. The data suggests that the overall increase in biomass is the dominant driver in increasing ET and thus ET/P. The authors argue that this "overrides physiological water saving responses". I do agree, however, the improved water use efficiency due to higher $CO_2$ levels might still be an important effect and could decrease ET, but clearly only for the same amount of biomass. For biomass aggregated results are presented, but there is no estimate of the physiological water saving response. Furthermore, the time series data does not provide statistical correlations between biomass and ET/P. Thus the study can not provide quantitative links between biomass or physiological water saving response and ET/P. However, both topics are suggested by the title and hypotheses. Therefore I recommend to adapt the red line of the manuscript or improve the analysis. Nevertheless, the observation that increases in forest biomass are potentially linked with increasing ET/P is important and should be communicated.

Response 1: We thank the reviewer for the suggestion of adjusting the red line of the manuscript in order to tone down statements of the role of the water saving response on evapotranspiration. This suggestion also agrees with that of Reviewer Nr. 1 that we also address in the Response 2 to that reviewer. Indeed, we have not quantified the stomatal water saving response, so the title, abstract and structure of the manuscript should be adjusted accordingly. We will do as such by removing the mention of the water saving response from the title and mentioning along the manuscript that our results of a consistent increase in the residual component of the evaporative ratio point to a dominating effect from an increase in forest biomass. Furthermore, we will mention in the Discussion that from such result we infer that a water saving response is either

weak or inexistent.

Concerning the suggestion to add a quantitative assessment, we have now included statistical correlations between the forest attributes of area, biomass, and composition (i.e., the ratio of leaf area index of deciduous forest to total leaf area index) and the residual component of the evaporative ratio $\Psi r = \Psi - \Psi c$ (See new Table 1 below). We have now added the calculation of the coefficient of determination ($R^2$) of the linear regression between all obtained annual values of the residual component of the evaporative ratio ($\Psi r = \Psi - \Psi c$) and the three mentioned attributes of forest structure for the temperate and boreal basin groups (forest cover, biomass and composition). The results agree with our previous results. We found that the $R^2$ for relationship between forest biomass and $\Psi r$ is significantly different from zero ($p<0.05$) for both the boreal and temperate groups and that forest biomass explains more of the variance of $\Psi r$ than the other forest attributes. In turn, $R^2$ for the relationship between forest cover and $\Psi r$ was only significantly different from zero ($p<0.05$) in the temperate group. Forest composition does not have any significant relationship with $\Psi r$ in any of the two basin groups, judging the low $R^2$ values and the high p-values. We are grateful to the reviewer for the suggestion to improve the analysis on this issue, since the addition of this new table will be a significant improvement of the study. We will include this new analysis, table and discussion in a revised manuscript.
* * *
New Table 1-

**BOREAL**

|  | biomass | forest cover | forest composition | fraction falling as snow |
|---|---|---|---|---|
|  | $V$ | $A$ | $Q_{LAI}$ | $f_s$ |
| intercept | -0.09 | 0.44 | -0.10 | 0.11 |
| slope | 0.00 | -0.78 | 0.89 | -0.15 |
| Adjusted $R^2$ | 0.07 | 0.02 | -0.02 | 0.02 |
| p-value | 0.028* | 0.168 | 0.629 | 0.187 |

**TEMPERATE**

|  | biomass | forest cover | forest composition | fraction falling as snow |
|---|---|---|---|---|
|  | $V$ | $A$ | $Q_{LAI}$ | $f_s$ |
| intercept | -0.06 | -0.25 | -0.16 | 0.02 |
| slope | 0.00 | 0.41 | 1.09 | -0.05 |
| Adjusted $R^2$ | 0.08 | 0.06 | 0.03 | -0.01 |
| p-value | 0.026* | 0.048* | 0.168 | 0.612 |

**Fig. 1.**

---

## Author Comment (AC5) · 18 Aug 2017

We thank the reviewers for their positive and constructive feedback to improve this manuscript. We appreciate that Reviewers 1 and 2 believe that our hypothesis and analysis are "well thought", "well executed" and "convincing". We also appreciate that Reviewers 1 and 4 think that the topic of the study is of "broad interest" and "well suited for the journal" and that our results are "important and should be communicated". The reviewers express different methodological concerns, in particular the somewhat negative Reviewer 3. In the author response to each of the reviews, we have addressed each of the questions, recommendations and concerns that they have very kindly posed.

In general, Reviewers 1, 2 and 4 have suggested toning down the statements regarding the tree water saving response (to increasing atmospheric $CO_2$ concentration) in the title, abstract and main body of the manuscript. We will do accordingly in a revised version. Furthermore, in order to address these reviewers concern, we have justified why we consider that the water saving response should still be present in the justification and discussion of a revised manuscript (see Response 3 to Reviewer 1). We will make sure that the overall message and conclusion is that increasing forest biomass due to forest management is an important driver of evapotranspiration change in this region. As the reviewers correctly remarked, our analysis does not explicitly detect or quantify the water saving response per se. What we discuss is rather that a possible $CO_2$-induced plant water saving response must be small or even inexistent compared to the result of this study which is a positive effect of increasing forest biomass (since we have a positive change in the residual component of the evaporative ratio over time). We will state this message more clearly in the revised manuscript.

Reviewers 2 and 4 have also suggested a statistical analysis to more quantitatively describe our results. We have incorporated such statistical analysis and included it in the response to the Reviewers. In summary, we calculated the coefficient of determination ($r^2$) of the linear regression between all obtained annual values of the residual component of the evaporative ratio ($\Psi r = \Psi - \Psi c$) and the annual values of the three mentioned attributes of forest structure for the temperate and boreal basin groups. The results of this statistical analysis confirm our previous conclusions: forest biomass (V) was the only forest structure attribute that could significantly ($p<0.05$) contribute to the variation in $\Psi r$ among years in both the boreal and temperate basin groups. In turn, forest cover (A) could only explain significantly ($p<0.05$) part of the variation in $\Psi r$ among

years in the temperate group, and forest composition could not significantly explain the variance in any of the two basin groups. We present these results in a new Table 1 as shown in the Author comments to the reviewers.

Reviewer 3 also thought that we should further explore if a change in the fraction of precipitation falling as snow (fs) had a significant effect on the residual of the evaporative ratio. In order to address this concern, we have also included this parameter in the statistical analysis previously described. We found that fs could not significantly explain the variation in $\Psi r$ among years in the two biomes. The reviewer has additionally requested an uncertainty analysis of our calculations. We have also done accordingly and shown in the Authors response to that reviewer with an modified Figure 6 that the incorporation of the uncertainty of precipitation (P), potential evapotranspiration (PET) and the landscape factor of the Chodbhury equation (n) does not modify our main finding of a general increase in the residual of the evaporative ratio in both basin groups during the period 1961-2012.

We hope that the new statistical analysis and uncertainty assessment previously described have increased the robustness of our analysis and conclusions and will satisfy the reviewers and editor. We also hope that after including all these updates into a revised manuscript as well as the other minor points on visualization, format and sentence structure, the Reviewers and the Editor consider that our manuscript is worth publication in HESS.

―――――――――――――――――

---

## Author Response (AR2)

**Second Response to Reviewers**

**Dominant effect of increasing forest biomass on evapotranspiration: Interpretations of movement in Budyko Space**

**Response to Reviewer Nr. 1**

**C1:** The authors gave a thorough revision in response to the previous reviewer comments. I have a few additional comments on the revised version:

With regard to R1, Q3, I still think there is a minor contradiction between the introduction (P2 L29-38) and the discussion (Section 4.2).

> 1. In the introduction, the authors present the stomatal water-saving response as a motivating hypothesis for their work and write "…it is indeed possible that there exists a water saving response…" in their reply. Then, in section 4.2, the authors use the same references to support their conclusion of a negligible water-saving response.

> 2. After reading the authors' references, my interpretation of the paragraph on P2 L29-38 is that no water-saving response is to be expected in this region for these species. The studies referenced by the authors in the introduction used a 90-100% increase in $CO_2$ concentration as compared to only a 27% actual increase over the period of the present study. The largest response of these studies was found by Rey and Jarvis (1998) – a 21% decrease in stomatal conductance for Silver Birch with a 100% increase of $CO_2$ concentration. Silver Birch on average make up less than 5% of the standing volume in the studied catchments. Therefore, a first guess would suggest a negligible decrease in stomatal conductance in the studied catchments.

> 3. My suggestion would be to revise the abstract, introduction, and section 4.2 to be clear that the water-saving response is an unlikely outcome for conifer-dominated forests that have experienced a $CO_2$ increase from 315 to 400 ppm. This would help the authors emphasize the more compelling motivation for their work, which is long-term water balance changes in catchments that have been managed for forestry, as well as their positive results.

**R1:** We thank the reviewer again for his valuable comments. In order to eliminate the minor contradiction mention by the Reviewer, we have changed the Introduction accordingly to state that the potential decrease in evapotranspiration resulting from $CO_2$-induced stomatal closure responses should not be expected to be as important in conifer-dominated forests as in broad-leaf dominated forests.

> *"Nordic experiments with Norway spruce showed no leaf water saving response to elevated CO2 (Hasper et al., 2016; Sigurdsson et al., 2013), while experiments with another conifer species, Scots pine, showed either a rather small reduction or no significant change in stomatal conductance under elevated CO2 (Kellomäki and Wang, 1996; Sigurdsson et al., 2002, respectively). In the deciduous species Silver birch, however, substantial CO2-induced leaf water saving responses were found (Rey and Jarvis, 1998)."*

Right after this, we now more clearly further justify one of the objectives of the study as follows:

> *"Stomatal water-saving responses may thus be expected to be small in conifer-dominated northern forests, but this remains to be tested at the basin level."*

*We have also modified the abstract to indicate that the stomata closure effect may be small in conifers if compared to that of angiosperms.*

> *"However, increasing global CO2 concentrations may also trigger physiological water saving responses in broadleaf tree species, and to a less degree in some needle-leaf conifer species, inducing an opposite effect."*

*Finally, we also simplified Section 4.2. of the Discussion to avoid repetition and redundancy of findings.*

**Q2:** Secondly, the end of the paragraph in section 4.3 is difficult to follow:

1. The authors reference a study (Zhang et al. 2015) and then state that its results do not apply to Swedish forests – I'd recommend deleting the reference or restating in a way that emphasizes its relevance to your study. It may not be needed given that Laudon et al. (2004, 2007) supports your point by itself.

2. On line 7, the authors state that the snow fraction may explain a small variance of (E/P)_r, but there is no reference to data that supports this statement.

3. How is spring snow related to the growing season length?

**R2:** Thanks for these thoughts. Based on this last remark, we have now removed the paragraph that was difficult to follow, since we now consider that it is rather speculative and does not contribute to the main findings of the manuscript. We also removed the reference of Zhang et al. (2015).

**Q3:** Lastly, Figure 7 needs y-axis labels. Also, it would help to reference panel 7c in the text where it is discussed (Page 7, Line 8).

**R3:** We fixed the y-axis labels and we also now reference panel 7c in the text.

**Response to Reviewer Nr. 2**

**Q1:** The authors have done an admirable job responding to my comments and those from the other reviewers and I thank them for their efforts. Most of my comments have been comprehensively addressed, and I have just one further minor comment about the new Figure 8. If I understand correctly, because there is a five year return interval on the plot data, using a 1- or 3-year associated averaging interval will change the specific inventory sites used to calculate V, A, and Qv, but the 5-year interval will always use all sites. If so, I recommend explicitly mentioning this in Section 2.5 for clarity. Furthermore, because the averaging is also being done on the annual evaporative radio residuals, I recommend re-naming the x-axis in Figure 8 something like "Averaging period".

**R1:** Thanks, we renamed the x-axis as "Number of years in averaged period" and we do the clarification suggested by the reviewer in L.4-5 of Page 6.

[revised manuscript text omitted]